

# Distribution of landfast, drift and glacier ice in Hornsund, Svalbard

Zuzanna M. Swirad[1], A. Malin Johansson[2], and Eirik Malnes[3]

[1]Institute of Geophysics, Polish Academy of Sciences, Warsaw, Poland
[2]Department of Physics and Technology, UiT The Arctic University of Norway, Tromsø, Norway
[3]NORCE Research AS, Oslo, Norway

*Correspondence: Zuzanna M. Swirad (zswirad@igf.edu.pl)*

**Abstract**. Co-occurring landfast, drift and glacier ice in fjords respond to climate differently and have diverse impacts on the environment. Here we describe a new method to separate ice types in fjord environments on 2639 binary satellite-derived ice/open water maps at 50 m resolution. We used a set of thresholds to create near-daily maps of landfast, drift and glacier ice of Hornsund, Svalbard over 11.5 years (2012-01-02 to 2023-06-29). The ice was first divided into stationary and moving classes based on ice-pixel persistence through time. The ice was then polygonised and the polygons were ascribed a set of parameters describing their class, time, size and location. Temporal and spatial constraints were imposed on landfast ice. Drift and glacier ice were split based on timing, location and size. Finally, the data were re-rasterized and refined at the pixel level. Over the 11.5 years, the fjord ice was classified as 53% drift, 35% as landfast, 8.5% as glacier, 1.4% as uncertain ice type while 2.1% was masked due to radar shadows. There was a great interannual variability in the length of sea ice and landfast ice seasons, and in ice type extent, with no clear long-term trend. Negative correlation existed between the water temperature in the winter months (January-March) and the length of the sea ice and landfast ice season, as well as between the air temperature in winter months and sea ice and landfast ice coverage. Glacier ice coverage depended on air temperature in summer months (July-September) and water temperature autumn months (October-December) where lower temperatures enhanced ice persistence. The method can be adapted to other areas and used in a wide range of analyses including fjord hydrography, nearshore wave transformation or ecological studies.

## 1 Introduction

Landfast ice, drift ice and glacier ice in fjords can be collectively referred to as fjord ice. The fjord ice has a great impact on e.g. energy, heat and light exchange between the atmosphere and the water, habitat conditions, type and efficiency of coastal processes, boat operation, and snowmobile transport (Nomura et al., 2018; Ricker et al., 2021; Loose et al., 2024). Sea ice is classified by the Global Climate Observing System (GCOS) as an Essential Climate Variable. Presence, extent and timing of the different components of the fjord ice depend on different aspects of the climate. For instance, colder air and water temperatures result in a larger extent of landfast ice (Muckenhuber et al., 2016). Abundance of drift ice is related to water temperature (de Steur et al., 2023) but its spatial extent may also link to ocean currents, wind direction and wind speed (Spreen et al., 2020; Mezzina et al., 2024; Muilwijk et al., 2024). Finally, higher water temperatures contribute to more intense tidewater glacier calving and increased amounts of glacier ice (ice melangé, brash ice, growlers, bergy bits, icebergs) in fjords (e.g.



Luckman et al., 2015; van Pelt et al., 2019). The different types of ice impact the environment differently as they are characterised by different salinity (sea vs freshwater), nutrient and sediment content (glacier ice brings material from terrestrial environment) and block shapes (sea ice for mammal resting spots, icebergs may be anchored to

the seabed at larger depths, etc).

There is a pertinent lack in a collective analysis of fjord ice which may partly result from canonical separation of sea ice and glacier ice mapping efforts, and differences in scales and detection methods. However, because of their cumulative contribution to the fjord environment as well as interactions and feedback, it is important to

approach fjord ice in a comprehensive way. Some examples of their interactions include in situ sea ice formation as glacier ice cools fjord waters (Styszyńska and Rozwadowska, 2008) or a decrease in glacier calving in times of high sea ice concentration (Pętlicki et al., 2015).

The automated mapping of ice extent in fjord environments using synthetic aperture radar (SAR) satellite imagery

has been limited, compared to the mapping efforts in the open ocean. The likely causes include the topography impact on radar shadowing, ocean surface wind patterns caused by tunnelling effects from the varying fjord side topography, mixed land/water pixels, tidal effect and presence of rocks and islands. Moreover, the different fjord ice types are characterised by similar radar backscatter values and, even if mapped collectively, the abundance of ice is difficult to relate to climate forcings (Swirad et al., 2024a).


In this study, we extended near-daily binary ice/open water maps of Swirad et al. (2023a) back in time to include three more sea ice seasons (Swirad et al, 2024b). We then used a set of calibrated thresholds to split the fjord ice into landfast, drift and glacier ice. The maps cover the southern part of Svalbard around the Polish Polar Station (PPS) Hornsund and expand over 11.5 seasonal cycles, 01.2012-06.2023. The length of the dataset combined with

extensive in situ measurements of air and water temperature made it possible to analyse seasonal and intern-annual trends and relate them to local climatological changes.

**2 Study area**

Hornsund is a ~300 km$^2$ fjord in south-western Spitsbergen, Svalbard. It has a ~12 km wide opening to the Greenland Sea. The fjord can be split into the main basin (~152 km$^2$) and the inner bays: Burgerbukta (~34 km$^2$),

Brepollen (~96 km$^2$) and Samarinvagen (~18 km$^2$). The average and maximum depths are ~100 and ~240 m, respectively (Fig. 1), with a tidal range of 0.75 m. The Sørkapp Current brings cold and fresh Arctic waters from the Barents Sea while the West Spitsbergen Current transports warm and saline Atlantic Water (Saloranta and Svendsen, 2001; Promińska et al., 2018).





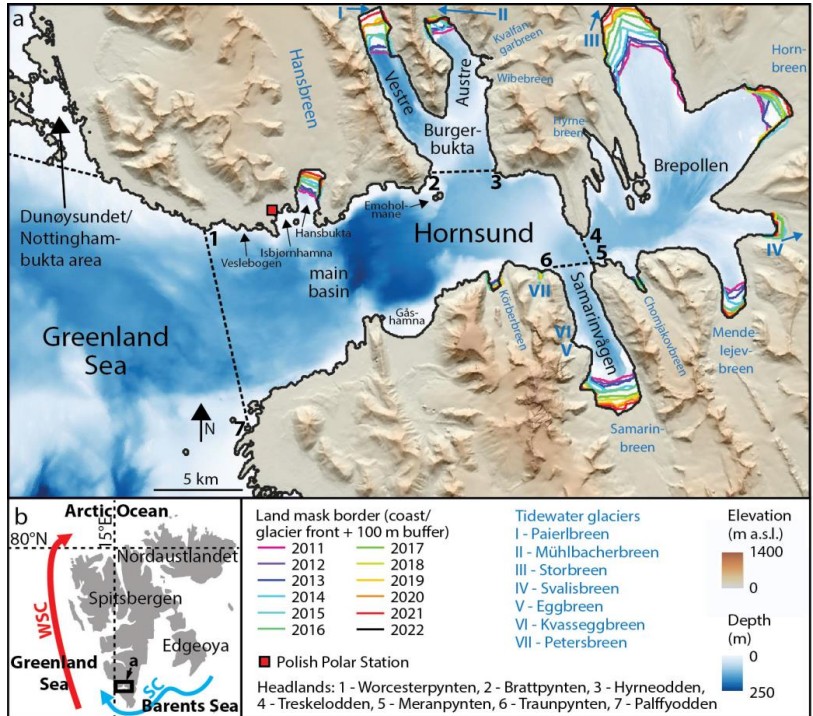

**Figure 1. Study area: a) Hornsund, b) Svalbard. WSC - West Spitsbergen Current. SC- Sørkapp Current. DEM courtesy of the Polar Geospatial Center; bathymetry IBCAO V5.0 Grid at 100 m and Norwegian Hydrographic Service at 5 m.**

The sea ice in Hornsund has a dual origin - it is either brought in from the Barent Sea with the Sørkapp Current or formed in situ by seawater freezing. Pack ice was present annually at the entrance to the fjord from October/December till May/July until 2005, when a regime shift occurred making it episodic in the Hornsund area (Herman et al., 2025). A decrease in landfast ice coverage after 2005 was also observed (Muckenhuber et al., 2016). Between 2014 and 2023 (nine sea ice seasons) the landfast ice started forming between December and March (average on February 4th) and disappeared between May and July (average on June 17th) (Swirad et al., 2024a). Spatially, higher and more consistent ice coverage characterised inner bays compared to the main basin. The highest coverage was in March in the main basin and in April in the bays, and there was a large interannual variability (Swirad et al., 2024a).

The mean daily air temperature at the PPS Hornsund weather station averaged -3.7°C in 1979-2018, but there has been an increase in mean annual air temperature by 1.14°C per decade with the largest increase in winter months. The easterly (from the inner fjord) winds dominated (mean direction 124°) and the mean wind speed was 5.5 m s$^{-1}$ (max 7.1 m s$^{-1}$ in February). Precipitation increased by 6.2 mm yr$^{-1}$ (Wawrzyniak and Osuch, 2020). Swell from south-southwest (open Greenland Sea) and locally generated short wind waves dominate with mean summer and winter offshore significant wave height of 1.2-1.3 and 2.6-2.7 m, respectively, which decrease to 0.2-0.4 and 0.3-0.5 m at the nearshore of the main basin (Wojtysiak et al., 2018; Herman et al., 2019; Swirad et al., 2023b). Pack



ice with the concentration >50% at the entrance to Hornsund effectively attenuates swell entering the fjord which results in lower significant wave height and longer wave period (Herman et al., 2025).

There are 16 tidewater glaciers in Hornsund which had the cumulative ice cliff length of 34.7 km in 2010. The glaciers retreated by 70 ± 15 m yr$^{-1}$ in 2001-2010 (Błaszczyk et al., 2013). Swirad et al. (2024a) observed a secondary peak in fjord ice coverage in October, ascribing it to the glacier calving.

## 3 Methods

### 3.1 Binary ice/open water classification

The ice/open water maps are based on RADARSAT-2 (RS-2) and Sentinel-1A/B (S-1) SAR scenes from Hornsund between January 2012 and June 2023. The S-1 dataset, processing workflow and validation against optical imagery were described in Swirad et al. (2024a) and here we only outline the RS-2 image processing.

### 3.1.1 RADARSAT-2 data processing

In total 797 RS-2 scenes fully covered the Hornsund fjord between 2012-01-02 and 2016-03-01, of which 638 (80%) were used for the ice mapping. Rejected scenes had either poor quality, lacked one of the polarization channels or occurred on the same day as a better quality scene. The remaining scenes had an average temporal frequency of 2.38 days for the entire period and 1.74 days until 2014-11-18 after which the acquisition became sporadic due to the S-1 data coverage over the area. The scenes were geocoded on a fixed grid with extent of X: 499-544 km and Y: 8531-8559 km (WGS84, UTM33N) and 50 × 50 m pixel spacing using GSAR software (Larsen et al., 2006). For each scene HH- and HV-channel radar backscatter sigma nought and incidence angle GeoTIFF rasters were created.

The Norwegian Polar Institute's land shapefile (NPI, 2014) was used to create a land mask that was updated annually using SAR image from July 1$^{st}$ (or the first available thereafter) to account for fjord area increase resulting from tidewater glacier retreat. Envisat ASAR scene was used for the 2011 mask, RS-2 for the 2012-2014 masks and S-1 for the subsequent masks. Each mask was applied on the July 1$^{st}$ - June 30$^{th}$ period. Additionally, a 100 m buffer was added along the coast to exclude the tidal zone.

Automated segmentation and classification followed the method developed by Cristea et al. (2020), and adapted to the Svalbard fjord environment by Johansson et al. (2020) and to Hornsund by Swirad et al. (2024a). A segmentation algorithm uses three raster layers: two polarization channels and an incidence angle layer. The algorithm considers the surface-specific intensity decay rate with increased incidence angle and it splits the image area into discontinuous segments using the traditional expectation-maximisation algorithm until the Pearson-style goodness-of-fit test is passed. Markov-random-field-based contextual smoothing and filtering were applied. The images were multi-looked 3 × 3 and log-transformed (Johansson et al., 2020). The number of segments ranged 3-6 with the majority (62%) of 5 segments. In 631 cases (99%) both HH and HV channels were used, five times only the HH channel and twice only the HV channel.



The 3-6 discontinuous segments were manually classified as 'ice' (1) or 'open water' (0). Of the 638 scenes, 499 (78%) did not require any modifications. For the remaining scenes the segments were polygonised in QGIS (v3.28) and some polygons were added or removed from the ice class (minor manual editing of 77 scenes, 12%) or entirely manually picked (62 scenes, 9.7%). The resulting binary maps are available in the PANGAEA repository (Swirad et al., 2024b) and accompanied by a document outlining the scenes and channels used, the number of segments and the level of manual editing.

### 3.1.2 RADARSAT-2 and Sentinel-1 data collation

On the top of the above described 638 binary maps based on RS-2 imagery, we used 2031 binary maps based on S-1 archive between 2014-10-14 and 2023-06-29 available in the PANGAEA repository (Swirad et al., 2023a).

Consistency in sea ice detection between different sensors was verified by Johansson et al. (2020) in Kongsfjorden and Rijpfjorden. Here, we used 29 pairs of binary maps available for the same day. The correlation coefficient for the ice extent from the RS-2 and S-1 image pairs is 0.96 and the RMSE is 52.6 km$^2$ (Fig. 2). The differences may be partly attributed to the ice drift and the time difference in image acquisition with RS-2 scenes taken either around 5:30-6:30 or 15:30-16:00 UTC and S-1 scenes taken around 6:00-6:30 UTC. 72% ice pixels overlap even though the ice drift makes pixel-by-pixel comparison difficult and underscores the performance. Slight differences in the detected sea ice may stem from a lower Noise Equivalent Sigma Zero (NESZ) for RS-2 (-26.5 to -31dB) compared to S-1 (-22dB) (https://sentiwiki.copernicus.eu/web/s1-mission; last access: 31 July 2025), making it comparably somewhat easier to identify newly formed sea ice in the RS-2 images and somewhat less sensitive to wet snow. Sight differences in the incidence angle between the images may also result in smaller sensor dependent differences in the binary maps.




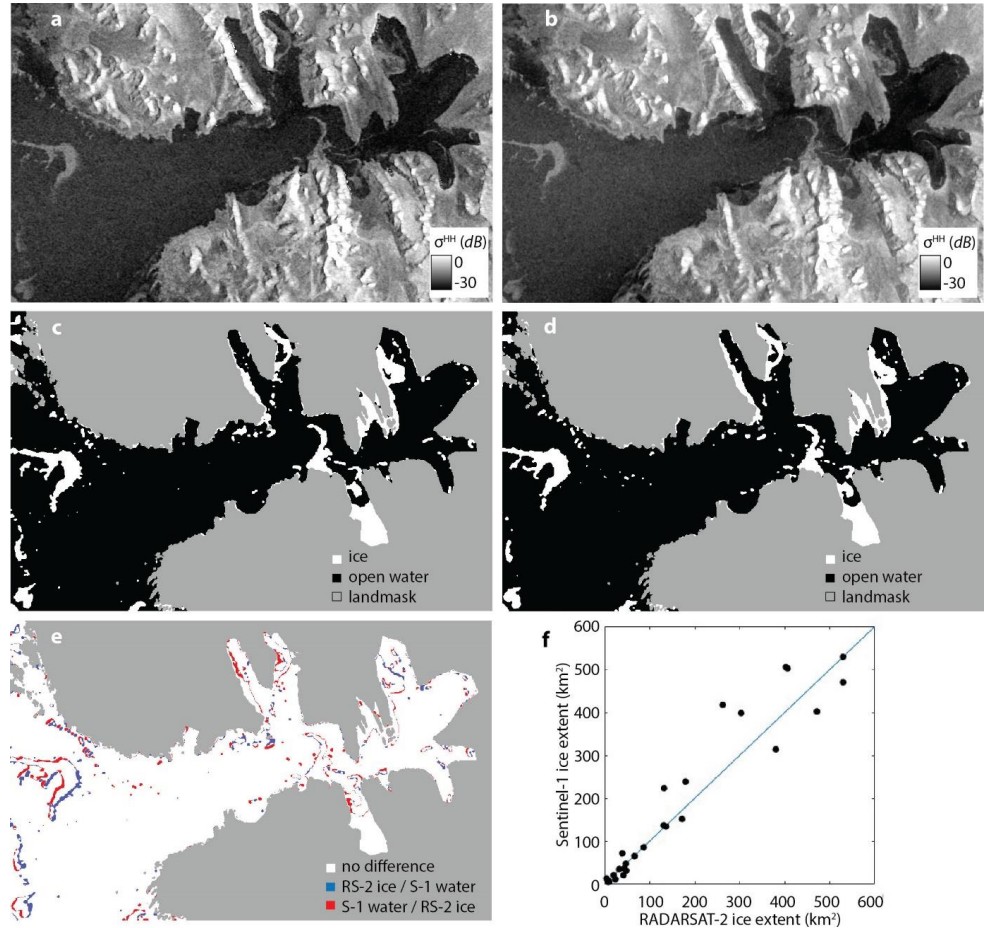

**Figure 2. Inter-sensor comparison of fjord ice mapping: a) RADARSAT-2 (RS-2) scene RS2_SCWA_20150105_053323_DES_152, b) Sentinel-1 (S-1) scene S1_EWM_20150105_054924_DES_110, c) RS-2 binary map, d) S-1 binary map, e) difference map, f) fjord ice extent for the 29 overlapping dates.**

The two datasets were combined into a single time series of 2639 ice maps. For the days when both RS-2 and S-1 maps were available, RS-2 scenes were used because of the lower noise floor (https://earth.esa.int/eogateway/documents/20142/0/Radarsat-2-Product-description.pdf; last access: 30 July 2025) and visual inspection of the scenes. The 2018-12-25 (S-1) map was omitted because of a size mismatch.

### 3.2 Separation of ice types

Although they are similar in backscatter values, we expected the types of ice (landfast, drift, glacier) to be separable by their persistence through time, location in the bay-sea domain, timing in the year, size and shape. For example, landfast ice is generally present in the bays over an extended time period, mostly in spring. Conversely, drift ice is mostly present in the open Greenland Sea and in the main basin of Hornsund, where its location changes at a daily to weekly scale and it is present mostly in winter and spring though it may occur at different periods too. Finally, glacier ice is predominantly present in summer and autumn (Luckman et al., 2015), mostly in the



bays though it sometimes drifts out to the main basin. Here we describe a workflow of separating types of fjord ice based on thresholds set based on the familiarity with the study area and understanding of the physical processes (Fig. 3).

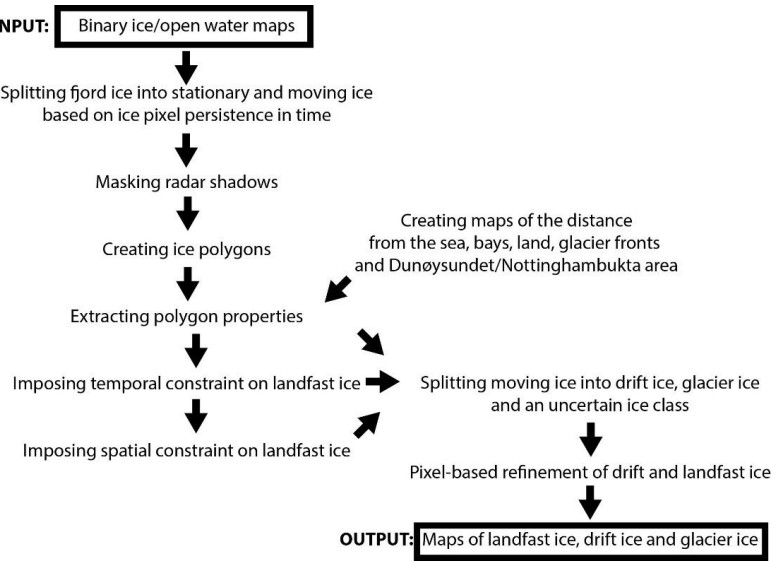

**Figure 3. Fjord ice type separation workflow.**


### 3.2.1 Fjord ice split into stationary and moving ice classes

The ice pixels were first divided into 'stationary' or 'moving' classes depending on their persistence through time. The aim of this step was to provide a rough separation of ice types under the assumption that landfast ice persists in the same locations longer than drift and glacier ice. To optimize the number of days for the separation, a 13-

month subset of maps (July 2018 - July 2019; $n$ = 293) was analysed. The number of cells ascribed 'stationary' and 'moving' was extracted for bays and open Greenland Sea (as delimited in Fig. 1) based on a persistence during 5, 10, 15 and 20 subsequent days. Fig. 4 shows that in the bays the number of moving pixels was dynamic throughout the year likely reflecting the glacier and drift ice fluxes as well as landfast ice edge fluctuations. Conversely the stationary ice was consistently absent during the summer and autumn months and present during

the late winter and spring months, with the exception of the 5-day persistence when ice appeared stationary in early autumn. In the open sea, the ice was present nearly exclusively in winter/spring. A 5- and 10-day persistence thresholds over this area meant that stationary ice was present despite the lack of locations for the landfast ice to form. We believe that these are the locations of recurrent presence of drift ice or where drift ice was temporarily halted. The 20-day threshold resulted in the highest amount of moving ice extent (Fig. 4), though the difference

to the 15-day threshold was limited. The 15-days threshold implies one to four repeat cycles for S-1, meaning that a range of incidence angles may be covered in each period. Based on these observations we set 15 days as a threshold for separating moving and stationary ice.





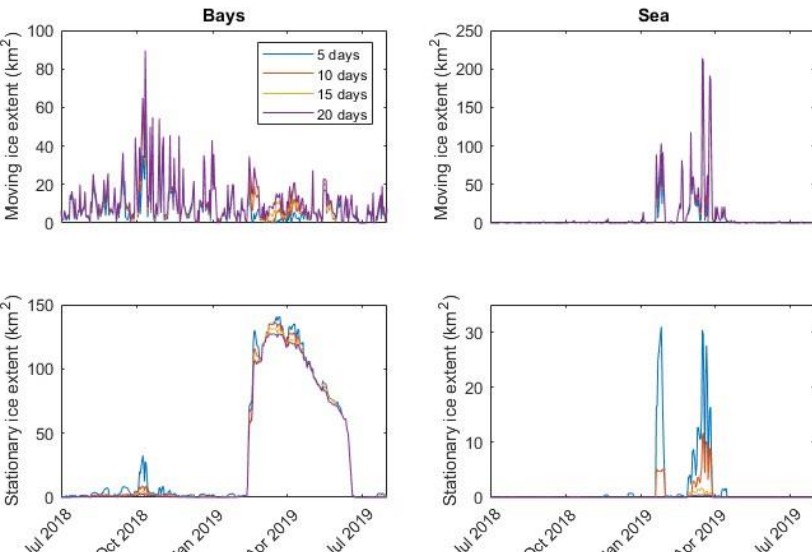

**Figure 4. The extent of moving and stationary ice between 2018-07-01 and 2019-07-31 depending on the minimal number of consecutive days of ice presence in the bays and open sea as delimited in Fig. 1.**

### 3.2.2 Radar shadow masks

Each orbit is characterised by a different area of uncertainty resulting from radar shadow and layover by topography. With the larger incidence angle the mountain shadows cover larger fjord areas. To mask out areas that may be erroneously classified as ice, radar masks were extracted for each used orbit separately ($n$ = 30 for RS-2 and $n$ = 11 for S-1). A 200 m buffer was applied around the shadowed zones (Fig. 5a-b). The radar masks were applied on the stationary/moving ice rasters, so that if ice pixels were within the masked area, they were given a separate class 'masked' and were not edited thereafter (Fig. 5c).

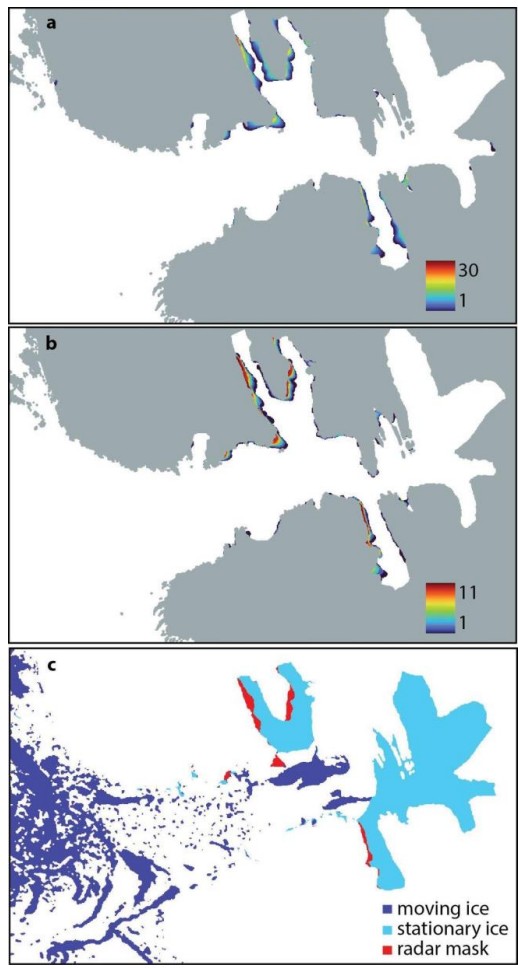


**Figure 5. Radar shadow masks: the number of orbits with a shadowed area for a) RADARSAT-2, b) Sentinel-1, c) an example radar mask applied on the 2013-04-29 split scene.**

### 3.2.3 Polygon classification

The annually-updated (July-June) maps of distance from land, bays and glacier fronts, and the single maps of distance from the open sea and to Dunøysundet/Nottinhambukta (DN) area were created in QGIS (Fig. 6). The masked moving and stationary ice maps were polygonised by the ice class so that all touching pixels of the same class were put into the same polygon. The polygons were characterised by a number of parameters: ice class, date, surface area, mean, minimal and maximal distance from the sea, bays, land, glacier fronts and the DN area, and

the minimal class value within a 50-m buffer (Table 1).



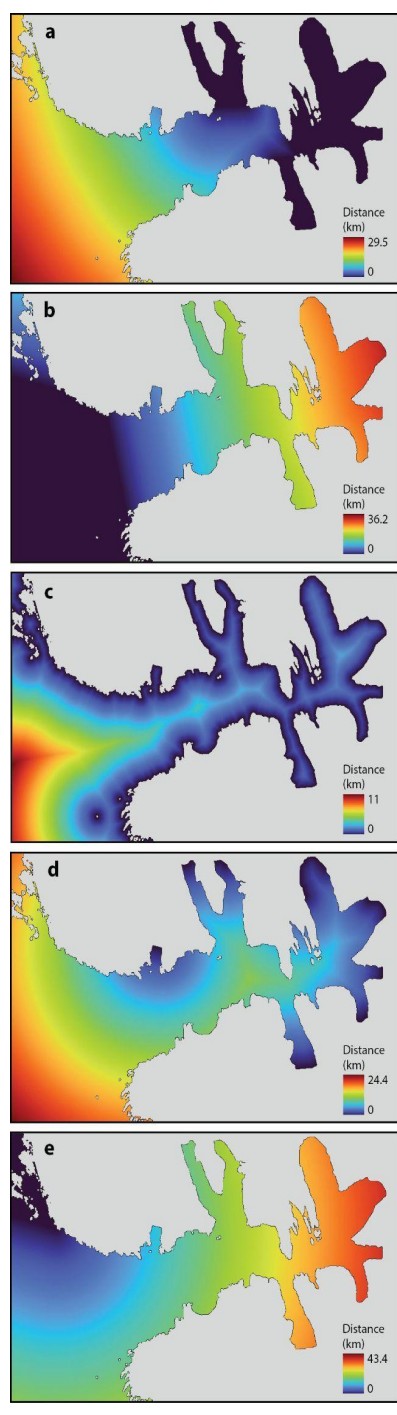

**Figure 6. Maps of the distance from: a) the bays, b) the sea, c) the land, d) the glacier fronts, and e) the Dunøysundet/Nottinghambukta (DN) area as delimited in Fig. 1 for July 2018 - June 2019.**




**Table 1. Parameters describing polygons for separating ice types.**

| Parameter (values/units) | Meaning/calculation | Expected importance | Stage of workflow |
|---|---|---|---|
| Ice class (1 = moving; 2 = stationary) | Stationary ice is persistent in a given location for at least 15 days. | | |
| Date (yyyy-mm-dd) | Extracted from the scene name | It determines whether the ice is in or outside the sea ice or landfast ice season (Table 2). | Temporal constraint on landfast ice; splitting moving ice |
| Surface area (m2) | QGIS Add Geometry Attributes tool | The smallest ice polygons may be speckle or ships; landfast ice polygon size is partly controlled by the bay size. | Spatial constraint on landfast ice; splitting moving ice |
| Distance from the bays (m) | Zonal statistics (mean, min, max) on the distance raster (Fig. 6a); varies every year | Landfast ice occurs in the bays. | Spatial constraint on landfast ice (min, max) |
| Distance from the sea (m) | Zonal statistics (mean, min, max) on the distance raster (Fig. 6b); the same for the whole dataset | Only drift ice is expected in the open sea. | Splitting moving ice (min) |
| Distance from the land (m) | Zonal statistics (mean, min, max) on the distance raster (Fig. 6c); varies every year | Ice polygons adjacent to the land may be waves breaking (white water) and rocks and shallows exposed during the low tide. | Spatial constraint on landfast ice (min, max); splitting moving ice (mean) |
| Distance from the glacier fronts (m) | Zonal statistics (mean, min, max) on the distance raster (Fig. 6d); varies every year | Glacier ice is located close to the glacier fronts. | Splitting moving ice (max) |
| Distance from the Dunøysundet/Nottinghambukta (DN) area (m) | Zonal statistics (mean, min, max) on the distance raster (Fig. 6e); the same for the whole dataset | Glacier ice occurs far from the DN area. | Splitting moving ice (min) |
| Minimal ice class value in a 50-m buffer (0 = no ice; 1 = moving) | Zonal statistics (min) on the raster with stationary, moving, masked ice and no ice (open water and landmask) | If the minimal value is '1' the polygon is 'moving ice' located inside a larger 'stationary ice' polygon and should be classified as 'landfast ice'. | Spatial constraint on landfast ice |

The importance of the inclusion of distance measures can be observed in Figure 7. The highest proportion of ice was present in the bays (A in Fig. 7a). The second peak of stationary ice in the open sea was likely caused by the continuous flow (re-occurrence of drift ice) or locally halted drift ice or, to a lesser extent, landfast ice presence

in the vicinity to the DN area (B in Fig. 7a). The intermediate peak for both ice classes (C in Fig. 7a) may indicate sheltered Hansbukta/Isbjørnhamna area and Gåshamna where landfast ice might have formed and drift ice could have persisted. In the former area the peak might also reflect glacier ice. The dominance of moving ice closer to the sea reflects the pack ice drifting from the Greenland Sea, while a higher proportion of stationary ice present at 15-20 km from the sea indicates the bays (D in Fig. 7b). Both ice classes occurred most often close to the bays

and to the land, which is not surprising given the sheltered character of these locations where the landfast ice could form and persist, drift ice could have been pushed from the main basin or formed from breaking up of the landfast ice, and glacier ice could have been produced (E in Fig. 7c). At glacier fronts located in the deepest parts of the bays favourable conditions existed for both the landfast ice to form and the glacier ice to persist. There is, however, quite a high variability in density distribution for various distances from the glacier fronts. This may be because

Hornsund bays are relatively large, and a sheltered area stretches far from the glacier fronts. For instance, the entrance to Brepollen, a place of abundant landfast and drift ice, is located 8-10 km from the glacier fronts (F in Fig. 7d). The peak at ~16 km from the glacier fronts indicates the DN zone (G in Fig. 7d). Finally, the DN area is also ice-rich with further peaks representing individual bays that are generally zones of both stationary and moving ice concentration (H in Fig. 7e).

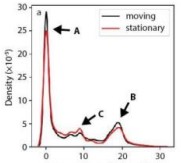 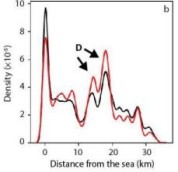 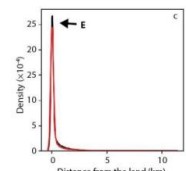 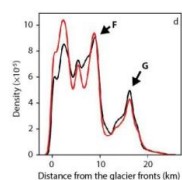 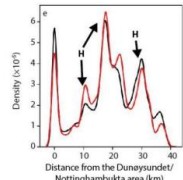


**Figure 7. Kernel density distribution of the mean distance from a) the bays, b) the sea, c) the land, d) the glacier fronts, and e) the Dunøysundet/Nottinghambukta (DN) area for the moving and stationary ice polygons in 2012-2023.**



The timeseries of the extent of the stationary and moving ice shows that the 15-day persistence threshold provided a rough separation between landfast (stationary) and drift/glacier (moving) ice (Fig. 8a). The consistent extent of stationary ice from late winter to the end of spring agrees with the timing of the landfast ice (Swirad et al., 2024a). The limited extent and duration of the landfast ice in winters with high air temperatures (2011/12 and 2013/14) are well captured (A in Fig. 8a). The moving ice in summer and autumn represents glacier ice (B in Fig. 8a). Drift ice is represented as the moving ice present during the entire winter and spring, slightly shifted relative to the landfast ice (C in Fig. 8a). This is consistent with the findings of Swirad et al. (2024a) who showed that the drift ice appeared on average 24 days before the landfast ice onset and was gone around 20 days before the end of the landfast ice season. During the seasons with low air temperatures (2012/13, 2014/15, 2019/20 and 2021/22) the persistence-based separation resulted in detecting stationary ice well-beyond the cumulative ~140 km$^2$ bay extent (D in Fig. 8a). A set of thresholds based on the familiarity with the study area and understanding of the physical processes was subsequently applied to improve the ice type separation.





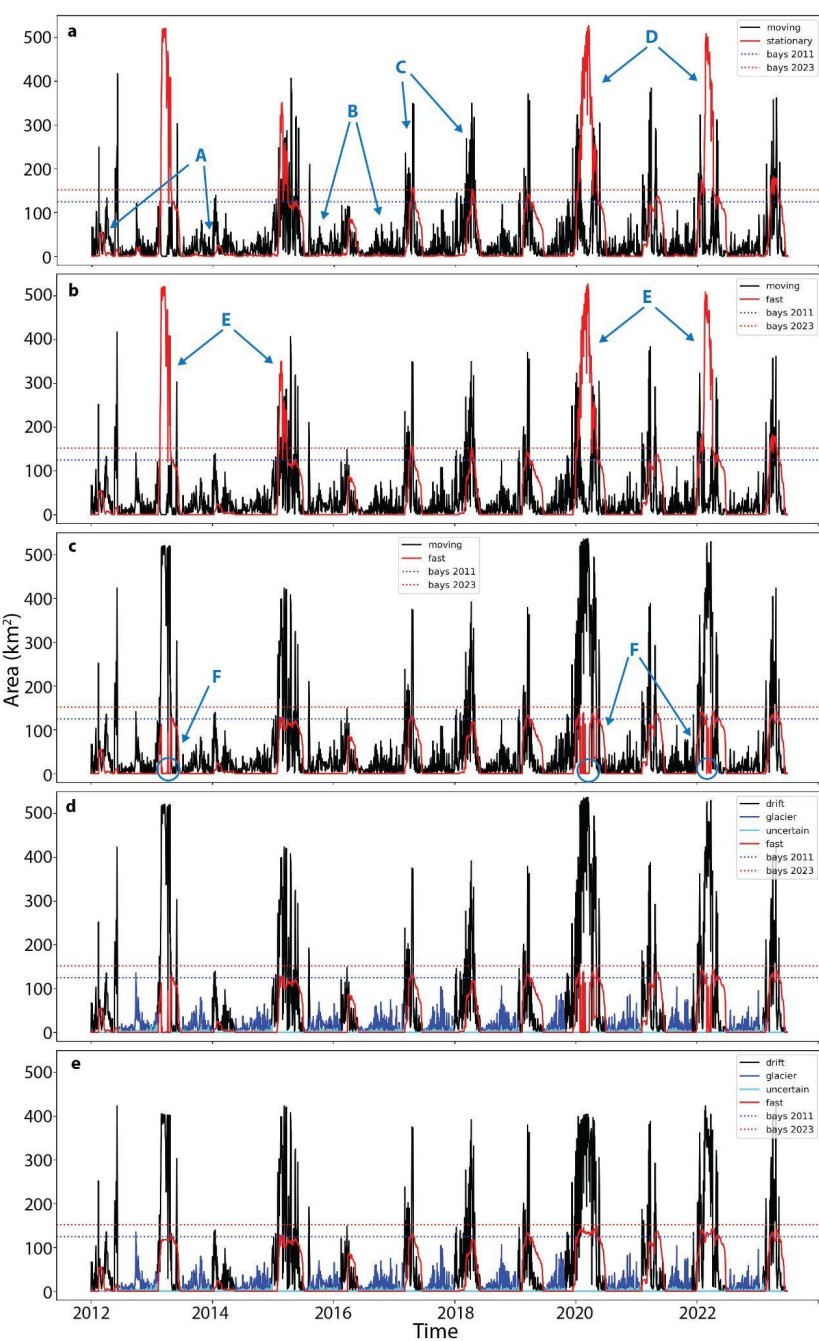

**Figure 8. Time series of ice classes and types in Hornsund in 2012-2023 at different stages of the workflow (Fig. 3): a) moving and stationary ice based on the 15-day persistence split, b) temporal constraint on the landfast ice, c) spatial constraint on the landfast ice, d) split of moving ice into drift and glacier ice, e) pixel-based refinement of drift and landfast ice. Note the bay increase over 12 years as the glaciers retreat.**




First, the landfast ice was extracted. The stationary ice that was outside the landfast ice season (Table 2) was put into the moving ice class. Then, the stationary ice that either was not adjacent to the land or stretched far from the bays was re-classified as moving ice. The landfast ice was allowed 1.5 km from the bays as it was observed to exceed Treskelodden and Traunpynten and surround Emoholmane islands. Small ($\leq 4$ km$^2$; dictated by the bay size) landfast ice polygons located no farther than 1.3 km from the land were also allowed at 5 to 9.5 km from the bays to encompass Hansbukta/Isbjørnhamna area and Gåshamna (for locations see Fig. 1). Moving ice fully surrounded by stationary ice was ascribed the landfast ice class. Figure 8b-c shows how the temporal and spatial thresholds helped delimit the landfast ice. During the cold seasons the temporal constraint still allowed the landfast ice to be present in the main basin and the open sea (E in Fig. 8b). The spatial constraint limited this effect but resulted in the entire polygon transition to the drift ice class (F in Fig. 8c). This was addressed later at the pixel level (last step in Fig. 3).

**Table 2. Timing and duration of the sea ice and landfast ice season in Hornsund from 2011-2023. The 2014-2023 data come from Swirad et al. (2024a).**

| Season | First drift ice | Landfast ice onset | Last drift ice | Landfast ice end | Length of sea ice season (days) | Length of landfast ice season (days) |
|---|---|---|---|---|---|---|
| 2011/12 | 11 Dec | 17 Feb | 06 Jun | 11 Jun | 184 | 116 |
| 2012/13 | 8 Feb | 11 Feb | 04 Jun | 21 Jun | 134 | 131 |
| 2013/14 | 2 Jan | 12 Jan | 09 Apr | 19 May | 138 | 128 |
| 2014/15 | 29 Dec | 27 Jan | 13 Jun | 07 Jul | 191 | 162 |
| 2015/16 | 15 Feb | 30 Mar | 26 May | 29 May | 105 | 61 |
| 2016/17 | 01 Mar | 07 Mar | 18 May | 18 Jun | 110 | 104 |
| 2017/18 | 30 Dec | 29 Jan | 23 May | 04 Jun | 157 | 127 |
| 2018/19 | 25 Jan | 12 Feb | 03 Jun | 18 Jun | 145 | 127 |
| 2019/20 | 28 Oct | 18 Dec | 26 May | 29 Jun | 246 | 195 |
| 2020/21 | 02 Feb | 04 Feb | 08 Jun | 24 Jun | 143 | 141 |
| 2021/22 | 03 Dec | 28 Dec | 23 May | 22 Jun | 202 | 177 |
| 2022/23 | 11 Feb | 21 Feb | 24 May | 12 Jun | 122 | 112 |
| **Average** | **11 Jan** | **4 Feb** | **26 May** | **16 Jun** | **156** | **132** |

Subsequently, a set of thresholds was set to split the moving ice into drift ice and glacier ice. An additional class called 'uncertain' was imposed on all objects classified as ice in the binary classification (Swirad et al., 2023a; 2024a; 2024b) which were not masked because of the radar shadows (Section 3.2.2) but did not fulfil the criteria to be classified as one of the three ice types. These locations may represent speckle noise resulting from low signal-to-noise ratio or from the atmospheric effect on backscatter variability but also ships, nearshore wave breaking (white water), or exposed rocks and shallows during low tide. The ice outside of the sea ice season was classified as glacier ice unless it was outside the fjord or was both > 3.5 km from the glacier front and focused along the coast (Table 2). The first condition was based on the field observations that the plumes of produced ice melangé, brash ice, growlers and bergy bits get dispersed and melt when still in the fjord while icebergs are rare in Hornsund and tend to be pushed nearshore and get anchored to the sea bottom rather than drift away to the open sea. The second condition is based on the observation that glacier ice persists nearshore close to the glacier fronts while the high backscatter values along the coast far from the glacier fronts likely represent white water and exposed rocks. Notably, the 100 m buffer around the land mask applied before running the segmentation algorithm (Section 3.1.1) made it impossible to detect the real ice accumulated at the shore.

During the sea ice season, the moving ice was considered drift ice. The motivation for this decision was the fact that there exists a high daily to weekly variability in drift ice distribution as well as the type of the drift ice which





includes the pack ice ranging from grey ice (10-15 cm) to thick first-year ice (< 2 m) and in situ ice broken from
the landfast ice (Swirad et al., 2024a). Drift ice may therefore be distributed throughout the study area and have a
full range of ice polygon morphology. Finally, drift ice was allowed outside of the sea ice season if it was located
in the open sea and was at least 1 km$^2$. The sea ice season over the Barents Sea is longer than that of Hornsund
and pack ice is observed sporadically in the southwestern part of the study area; usually it happens in late autumn
- early winter before the sea ice season in Hornsund starts, but may happen earlier too with the high sea ice
concentration in the fjord e.g. in August 2015. Ignoring the small polygons decreases the chance of misclassifying
speckle noise, ships, white water and exposed rocks as drift ice.

### 3.2.4 Pixel-based refinement

Finally, the different ice type polygons and no ice zones were rasterized back into the maps with the extent and
resolution of the original binary maps. Time series of pixel values were used to re-classify drift ice into landfast
ice in bays in the periods of high sea ice coverage over the entire fjord (Fig. 8e). Table 3 contains the information
on the changes of spatial extent of different ice types at various stages of the workflow. The ice type maps were
visually inspected.

**Table 3. Average spatial extent (km$^2$) of the ice types at various stages of the workflow.**

| Stage | Stationary | Moving | Landfast | Drift | Glacier | Uncertain | Masked |
|---|---|---|---|---|---|---|---|
| Before thresholding | 46.20 | 37.31 | - | - | - | - | 1.83 |
| Temporal constraint on landfast ice | - | 38.00 | 45.51 | - | - | - | 1.83 |
| Spatial constraint on landfast ice | - | 57.92 | 25.59 | - | - | - | 1.83 |
| Split of moving ice | - | - | 25.59 | 49.49 | 7.21 | 1.22 | 1.83 |
| Pixel-based refinement of drift and landfast ice | - | - | 30.19 | 44.89 | 7.21 | 1.22 | 1.83 |


### 3.3 Environmental datasets

Mean daily air temperature for the period 2011-07-01 to 2023-06-30 derived from 3-hourly automatic
measurements with Vaisala HMP45D and HMP155 probes at the PPS Hornsund were obtained from Wawrzyniak
and Osuch (2020) and the PPS archive (https://monitoring-hornsund.igf.edu.pl/index.php/login; last access: 2025-
07-21). The probes are located in the vicinity of the main station building, ca. 200 m from the shore, at an elevation
of 2 m a.g.l, which is ~12 m a.s.l.

Hourly water temperature from the sea bottom moorings were collected as part of the LONGHORN oceanographic
monitoring of the PPS (https://dataportal.igf.edu.pl/dataset/temp_sal_one_hour_averaged_mooring_data; last
access: 2025-07-21). The data from eight deployments spanning 2014-06-01 to 2024-06-09 were used that
included nearshore (10-23 m depth) locations in the north-western Hornsund (Hansbukta, Isbjørnhamna and
Veslebogen; Fig. 1). The measurements were taken with HOBO Water Temp Pro v2 (U22-001), RBRsolo T and
RBRconcerto CTD (Table 4). Daily water temperature was calculated by averaging over the hourly data.




**Table 4. Water temperature data used in the study.**

| LONGHORN ID | Latitude (°N) | Longitude (°E) | Depth (m) | Start | End | Days | Instrument |
|---|---|---|---|---|---|---|---|
| T01 | 77.0009 | 15.6246 | 23 | 01.06.2014 | 21.05.2015 | 355 | HOBO Water Temp Pro v2 (U22-001) |
| T03 | 77.0031 | 15.6298 | 22 | 10.06.2015 | 26.06.2016 | 383 | HOBO Water Temp Pro v2 (U22-001) |
| T06 | 76.9977 | 15.5605 | 10 | 13.06.2016 | 23.05.2017 | 345 | RBRsolo T |
| T07 | 76.9977 | 15.5599 | 10 | 02.06.2017 | 22.05.2018 | 355 | RBRsolo T |
| CTD03 | 76.9951 | 15.4881 | 17 | 05.06.2018 | 09.06.2019 | 370 | RBRconcerto CTD |
| CTD04 | 76.9952 | 15.4894 | 16 | 25.06.2019 | 05.03.2021 | 620 | RBRconcerto CTD |
| CTD07 | 77.0029 | 15.5843 | 11 | 08.09.2022 | 02.06.2023 | 268 | RBRconcerto CTD |
| CTD09 | 76.9955 | 15.5665 | 16 | 11.06.2023 | 09.06.2024 | 365 | RBRconcerto CTD |

Finally, the landmasks (Section 3.1.1) were used to calculate the gain of fjord surface area and approximate
tidewater glacier front retreat.

**4 Results**

Ice type maps at 50 m resolution spanning from 2012-01-02 to 2023-06-29 ($n$ = 2639; ~1.59 days map$^{-1}$) include
seven classes: open water (0), drift ice (1), landfast ice (2), glacier ice (3), uncertain (7), masked (8) and landmask
(9) (Swirad, 2025). The uncertain and masked classes were classified as ice in the binary maps (Swirad et al.,
2023a; 2024b) but were disregarded from the main ice type classes because of the radar shadows (masked class)
or not fulfilling the ice type criteria and likely representing speckle, white water, ships etc (uncertain class).

Averaged over the 11.5-year study period, 53% of the ice was classified as drift, 35% as landfast, 8.5% as glacier,
2.1% as masked and 1.4% as uncertain (Table 3). The abundance and location of the different ice types varied
during the monitoring period. Drift ice had the largest extent that included the open sea, DN area, main basin and
the bays. It reached the maximum average 28% presence in the DN area. Landfast ice concentrated in the bays
with the most frequent appearance (up to 33%) in the deepest parts of Samarinvågen and Brepollen. Glacier ice
persisted in bays of tidewater glaciers with the highest recurrence (16%) in Vestre Burgerbukta where Paierlbreen
is located. The uncertain class dominated in the open Greenland Sea (speckle) and nearshore (white water, tidal
zone). Masked areas mostly covered western parts of Burgerbukta and Samarinvågen characterised by steep
mountain slopes (Fig. 9).




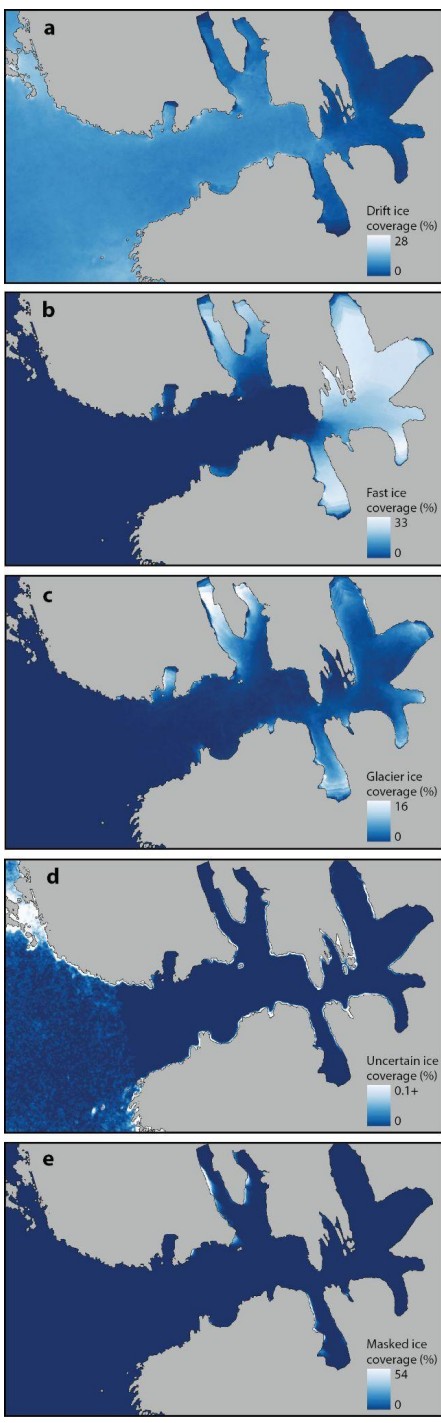

**Figure 9. Average ice presence in Hornsund between 2012-01-02 and 2023-06-29: a) landfast ice, b) drift ice, c) glacier ice, d) uncertain ice, and e) masked ice.**





The study period included 12 sea ice seasons (2011/12-2022/23) and 11 summer periods (2012-2022). The ice pack from the Barent Sea appeared between Oct 28th (2019/20) and Mar 1st (2016/17) with an average on Jan 11th. The landfast ice season started between Dec 18th (2019/20) and Mar 30th (2015/16) with an average on Feb 4th. The length of sea ice seasons varied from 105 days in 1015/16 to 246 days in 2019/20 (average 156 days). The landfast ice season varied from 61 days in 2015/16 to 195 days in 2019/20 (average 132 days) (Table 2). The least

icy seasons were 2013/14, 2011/12 and 2015/16, while the iciest were 2019/20 and 2021/22 (Fig. 10).

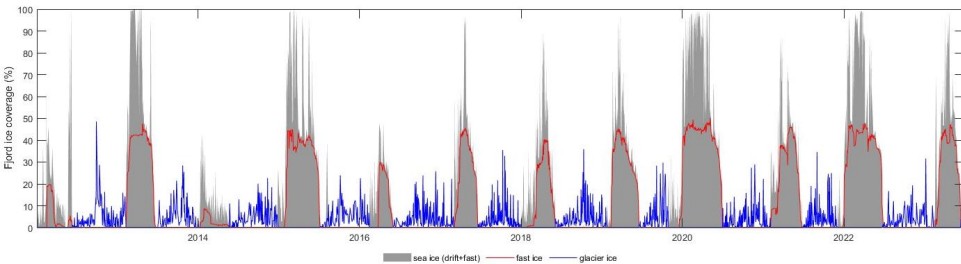

**Figure 10. Time series of fjord ice coverage in Hornsund from 2012-2023.**

On average, sea ice (drift and landfast ice) covered 40% of the fjord area during the sea ice season with the highest

coverage in 2012/13, 2019/20 and 2021/22, and the lowest in 2013/14 and 2011/12. The landfast ice covered on average 26% of the fjord area during the landfast ice season, but it ranged from 3.1% in 2013/14 and 4.6% in 2011/12 to 40% in 2019/20 and 39% in 2021/22. Glacier ice coverage during the sea-ice-free seasons averaged 4.1% and ranged from 3.7% in 2020 to 5.2% in 2015 (Table 5). The seasons 2011/12 and 2013/14 were characterised by the lowest mean coverage and smallest spatial extent of the landfast ice that did not encompass

all legs of the bays (Fig. 11). Glacier ice was consistently present in Burgerbukta and Samarinvågen but for Hansbreen and the Brepollen glaciers it was more diverse from year to year (Fig. 12).

**Table 5. Average seasonal ice type extent and coverage in Hornsund from 2011-2023. For the dates see Table 2.**

| Season | Sea ice (drift+landfast) extent during the sea ice season (km2) | Sea ice (drift+landfast) ice coverage during the sea ice season (%) | Drift ice extent during the sea ice season (km²) | Drift ice coverage during the sea ice season (%) | Landfast ice extent during the landfast ice season (km²) | Landfast ice extent during the landfast ice season (%) | Glacier ice extent during the preceding sea-ice-free season (km²) | Glacier ice coverage during the preceding sea-ice-free season (%) |
|---|---|---|---|---|---|---|---|---|
| 2011/12 | 47.01 | 17.07 | 38.17 | 13.86 | 12.55 | 4.56 | - | - |
| 2012/13 | 180.17 | 64.75 | 72.11 | 25.92 | 98.62 | 35.45 | 11.76 | 4.23 |
| 2013/14 | 37.80 | 13.48 | 28.53 | 10.18 | 8.63 | 3.08 | 13.88 | 4.95 |
| 2014/15 | 144.64 | 50.53 | 56.21 | 19.64 | 99.02 | 34.59 | 13.92 | 4.87 |
| 2015/16 | 60.60 | 21.01 | 26.93 | 9.33 | 59.91 | 20.77 | 15.12 | 5.24 |
| 2016/17 | 131.66 | 45.59 | 49.62 | 17.18 | 83.52 | 28.92 | 12.45 | 4.31 |
| 2017/18 | 84.99 | 28.97 | 39.82 | 13.57 | 54.45 | 18.56 | 11.64 | 3.97 |
| 2018/19 | 120.11 | 40.75 | 31.36 | 10.64 | 98.64 | 33.46 | 13.42 | 4.55 |
| 2019/20 | 173.11 | 58.47 | 77.68 | 26.24 | 118.31 | 39.96 | 14.90 | 5.03 |
| 2020/21 | 121.38 | 40.91 | 36.30 | 12.24 | 80.03 | 26.97 | 11.04 | 3.72 |
| 2021/22 | 158.70 | 53.10 | 65.21 | 21.82 | 115.31 | 38.58 | 13.51 | 4.52 |
| 2022/23 | 145.79 | 48.53 | 50.85 | 16.93 | 97.09 | 32.32 | 12.35 | 4.11 |
| **Average** | **117.16** | **40.26** | **47.73** | **16.46** | **77.17** | **26.43** | **12.00** | **4.13** |



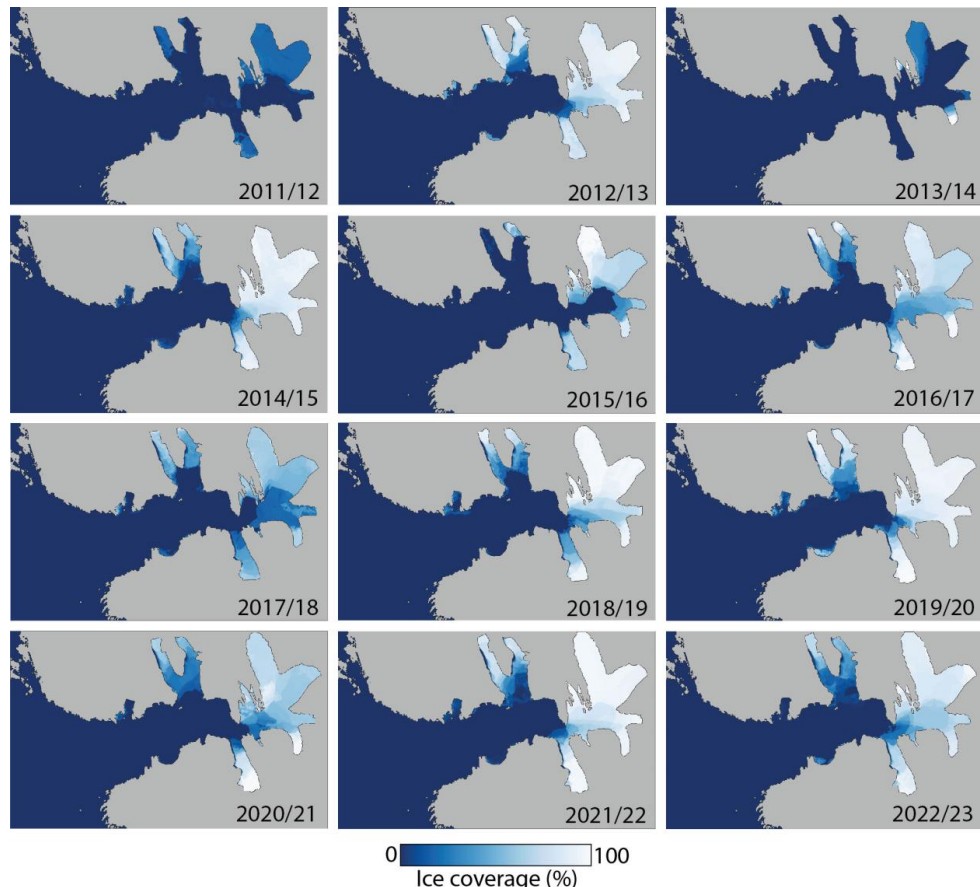


**Figure 11. Average landfast ice presence in Hornsund for the landfast ice seasons 2011/12 to 2022/23 (for dates see Table 2).**



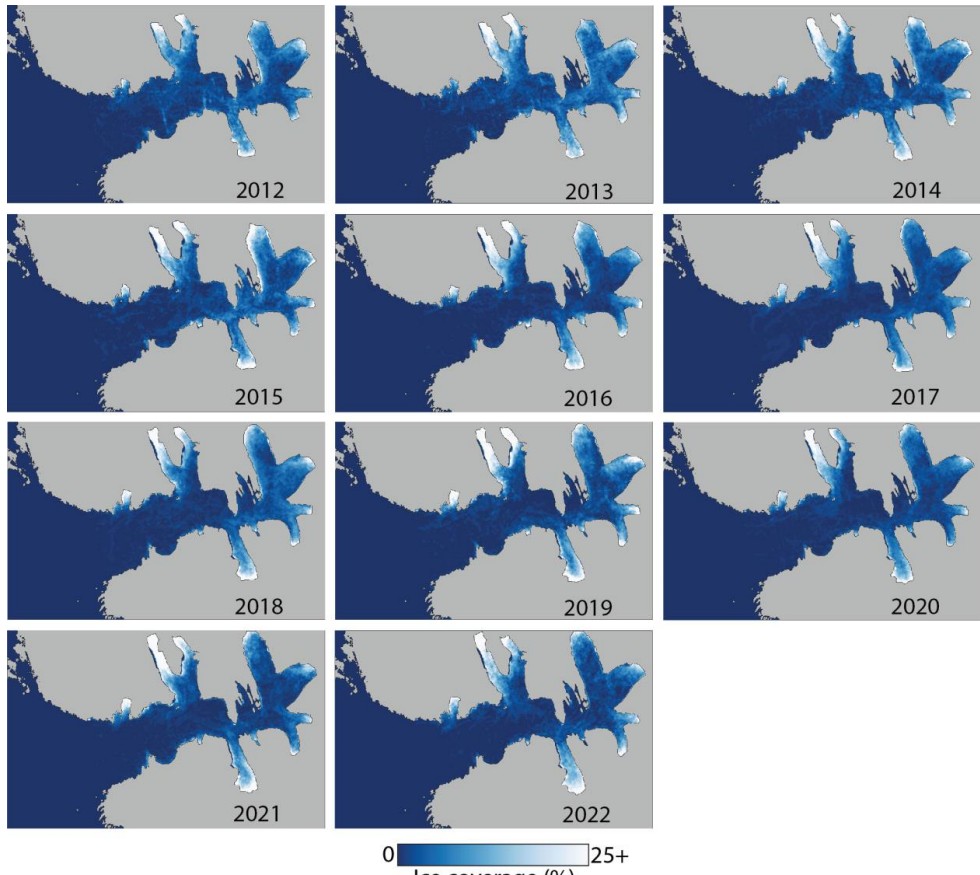

**Figure 12. Average glacier ice presence in Hornsund for the sea-ice-free seasons 2012 to 2022 (for dates see Table 2).**

The ice type distribution also varied seasonally. Drift ice had the highest coverage in March (25%) with up to 53% in 2012/13, 47% in 2019/20 and 46% in 2021/22. Landfast ice had the highest coverage in April (33%), but in 2019/20 the coverage over 40% persisted for five and in 2021/22 for four months. The highest coverage of

glacier ice characterised October (7.9%) (Fig. 13).



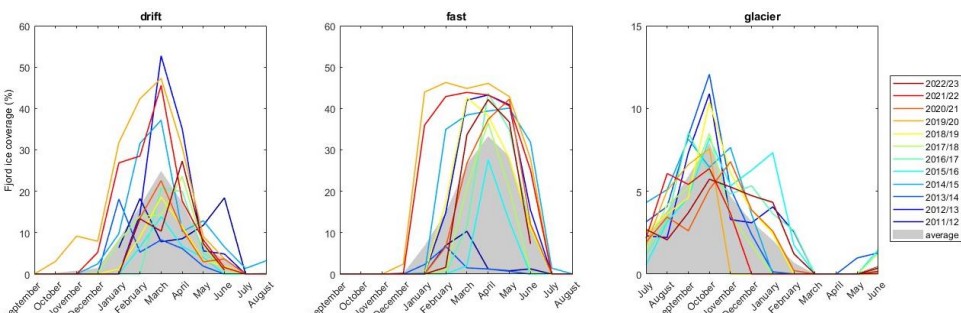

**Figure 13. Mean monthly extent of fjord ice in Hornsund from 2012-2023: a) drift ice, b) landfast ice, c) glacier ice. Note that both axes differ for the glacier ice.**


The timeseries of air and water temperatures show a clear seasonal pattern with higher temperatures in summer and lower in winter. The mean air temperatures in summer months (July-September) were relatively similar from year to year (4.0 to 5.4°C), while the largest variability characterised winter (January-March; -12.8 to -3.0°C) and autumn (October-December; -5.8 to 0°C), followed by spring (April-June; -2.6 to -0.1°C). The coldest seasons

were 2019/20 and 2012/13. Cold autumns generally did not associate with cold winters, except for the coldest 2019/20. Water temperatures were lower than air temperatures during the summer months, and higher during the rest of the year. The 2019/20 season was the only one with negative mean autumn temperatures, while 2019/20 and 2013/14 had -1.9°C mean in winter (Fig.14; Table 6).

**Figure 14. Time series of daily air and water temperature in Hornsund between 2011-07-01 and 2023-06-30.**





**Table 6. Seasonal variability in air and water temperature in Hornsund from 2011-2023.**

| Season | Mean air temperature in Jul-Sep (°C) | Mean air temperature in Oct-Dec (°C) | Mean air temperature in Jan-Mar (°C) | Mean air temperature in Apr-Jun (°C) | Mean water temperature in Jul-Sep (°C) | Mean water temperature in Oct-Dec (°C) | Mean water temperature in Jan-Mar (°C) | Mean water temperature in Apr-Jun (°C) |
|---|---|---|---|---|---|---|---|---|
| 2011/12 | 4.06 | -3.09 | -2.99 | -2.54 | - | - | - | - |
| 2012/13 | 4.07 | -3.08 | -9.46 | -2.45 | - | - | - | - |
| 2013/14 | 5.43 | -4.93 | -3.56 | -1.96 | - | - | - | - |
| 2014/15 | 4.31 | -4.10 | -7.47 | -1.39 | 4.24 | 0.63 | -1.92 | -1.21 |
| 2015/16 | 4.32 | -2.41 | -3.93 | -0.07 | 2.14 | 0.29 | -0.59 | -0.15 |
| 2016/17 | 5.06 | -0.04 | -7.53 | -2.60 | 3.47 | 2.34 | -0.89 | -0.45 |
| 2017/18 | 5.13 | -1.54 | -6.27 | -1.03 | 3.69 | 1.21 | -1.57 | -0.14 |
| 2018/19 | 4.01 | -2.24 | -8.95 | -0.40 | 3.26 | 1.24 | -1.44 | 0.20 |
| 2019/20 | 4.19 | -5.83 | -12.82 | -1.73 | 3.49 | -0.71 | -1.90 | -0.65 |
| 2020/21 | 4.79 | -2.33 | -6.34 | -1.81 | 3.47 | 1.02 | -1.28 | - |
| 2021/22 | 4.26 | -4.57 | -8.47 | -2.38 | - | - | - | - |
| 2022/23 | 4.87 | -2.28 | -6.25 | -0.44 | - | 1.10 | -1.23 | -0.09 |
| **Average** | **4.54** | **-3.04** | **-7.00** | **-1.57** | **3.39** | **0.89** | **-1.35** | **-0.35** |

The yearly updated landmasks reflected the result of glacier retreat on study area (fjord water) increase over the monitored 12 seasons, with a total increase of 29.76 km$^2$, equivalent to 11% relative to 2011 or 2.48 km$^2$ yr$^{-1}$. The largest area gain was for Brepollen (16.87 km$^2$; 21%) but in Samarinvågen which gained three times less area, the Samarinbreen retreat contributed to a 39% gain. The main basin experienced the lowest area increase (2.4 km$^2$; 1.6%) due to the Hansbreen retreat. The gain was not uniform through time with the highest fjord area increase in
2013/14 (5.87 km$^2$), 2022/23 (4.81 km$^2$) and 2016/17 (4.55 km$^2$) and the lowest in 2015/16 (0.35 km$^2$), 2019/20 (0.65 km$^2$) and 2018/19 (1.28 km$^2$) (Table 7).

**Table 7. Surface area of Hornsund and its parts (km$^2$) as delimited in Fig. 1 and the annual fjord area gain based on the 2011-2023 land masks. The landmasks were based on NPI (2014), updated annually with a SAR image from July**
**1$^{st}$ (or the first available thereafter) and a 100 m buffer was added along the coast to exclude the tidal zone.**

| Land mask | Hornsund | Main basin | Burgerbukta | Brepollen | Samarinvagen | All bays | Annual gain |
|---|---|---|---|---|---|---|---|
| 2011 | 275.46 | 150.74 | 29.81 | 81.91 | 13.01 | 124.73 | - |
| 2012 | 278.23 | 150.90 | 30.45 | 83.49 | 13.40 | 127.34 | 2.77 |
| 2013 | 280.38 | 151.09 | 30.69 | 84.98 | 13.62 | 129.29 | 2.15 |
| 2014 | 286.25 | 151.86 | 31.50 | 88.57 | 14.33 | 134.39 | 5.87 |
| 2015 | 288.47 | 151.69 | 32.02 | 90.05 | 14.71 | 136.79 | 2.23 |
| 2016 | 288.82 | 151.68 | 31.87 | 90.39 | 14.88 | 137.14 | 0.35 |
| 2017 | 293.37 | 151.93 | 32.96 | 92.72 | 15.76 | 141.44 | 4.55 |
| 2018 | 294.78 | 152.07 | 33.20 | 93.36 | 16.17 | 142.72 | 1.41 |
| 2019 | 296.07 | 152.11 | 33.15 | 94.16 | 16.64 | 143.95 | 1.28 |
| 2020 | 296.72 | 152.07 | 33.13 | 94.66 | 16.86 | 144.65 | 0.65 |
| 2021 | 298.84 | 152.24 | 33.85 | 95.47 | 17.28 | 146.60 | 2.12 |
| 2022 | 300.40 | 152.57 | 34.14 | 95.99 | 17.71 | 147.83 | 1.57 |
| 2023 | 305.22 | 153.14 | 35.20 | 98.78 | 18.09 | 152.08 | 4.81 |
| **Average** | **291.00** | **151.85** | **32.46** | **91.12** | **15.57** | **139.15** | **2.48** |

The length of sea and landfast ice seasons, and the coverage of sea, drift and landfast ice were positively correlated with one another with the highest correlation of the length of sea ice seasons versus the length of landfast ice season (Pearson correlation coefficient, PCC = 0.95) and of the sea ice coverage versus landfast ice coverage
(PCC = 0.92). The sea ice parameters did not correlate with the glacier ice coverage in the preceding sea-ice-free period. The length of the ice season was negatively correlated with the air and water temperatures with the strongest correlation between the water temperature in winter months and the length of the landfast ice season (PCC = -0.94) and the sea ice season (PCC = -0.89). However, the winter air rather than water temperature was more important for the coverage of sea ice (PCC = -0.87), landfast ice (PCC = -0.86) and drift ice (PCC = -0.78).
The air and water temperatures themselves were usually non-correlated except for a strong positive correlation in autumn (PCC = 0.92) and a moderate correlation of autumn and winter air temperatures versus winter water



temperatures (PCC = 0.67). Glacier ice coverage had a negative correlation with summer air and autumn water temperatures (PCC = -0.75 and -0.72, respectively). Fjord area gain was correlated positively to autumn water temperatures (PCC = 0.8) and negatively to spring air temperatures (PCC = -0.78) (Fig. 15).

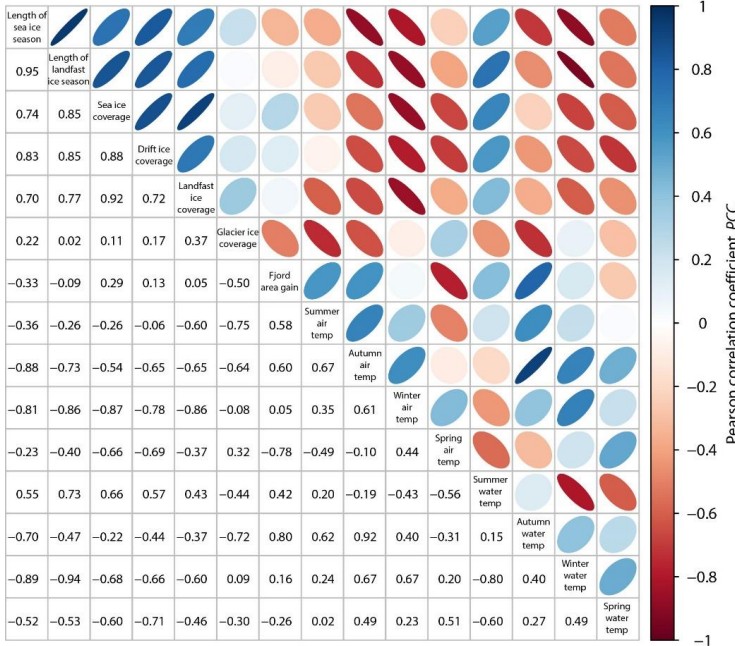

**Figure 15. Relationships between ice parameters and environmental forcings. Temperatures refer to the mean monthly for July-September (summer), October-December (autumn), January-March (winter) and April-June (spring). For the values and units see Tables 2, 5, 6 and 7.**

## 5 Discussion

Automated fjord ice mapping, and more widely nearshore ice mapping has been rarely attempted likely due to the challenges related to the mixed land/water pixels, presence of islands, rock and shallows, wave breaking and impact of topography on sea-state and radar shadowing. In addition, the often sheltered areas present in Hornsund result in level sea ice formation and a lack of textural features often used for sea ice classification (e.g., Lohse et al., 2019; Guo et al., 2025). Moreover, the level ice and an early ice formation have at times low signal-to-noise ratio (SNR), making accurate ice type classification challenging (Park et al., 2020). Here we followed the method of binary ice/open water mapping developed by Cristea et al. (2020) for open ocean, and adapted to the Svalbard fjords by Johansson et al. (2020) and to Hornsund specifically by Swirad et al. (2024a). Limitations of the method were discussed by Swirad et al. (2024a).

To our knowledge, this is the first attempt to automatically separate ice types in a fjord. Splitting ice types using only radar backscatter sigma nought values and the incidence angle is challenging given similar backscatter values for the various ice types, and the varying wind conditions due to the topography and coastal areas. It might be possible to use the backscatter values to investigate properties of the sea ice affecting the backscatter, such as early ice thickness growth, surface roughness, changes to the ice snow cover due to melting and wind effects. In




situ observations of snow cover made by the PPS Hornsund crew, as well as the weather station and the ocean
       mooring system enable robust assessments of such changes. Moreover, InSAR can be used to identify the landfast
       ice extent provided sufficient SNR values, i.e. thickness dependent due to the level ice formation (Dierking et al.,
       2017). However, we believe that in order to divide fjord ice according to its origin, the threshold-based method
       described in here is the most suitable. The method is simple, computationally cheap and it is based on the

familiarity with the study site and understanding of ice processes. It uses a set of thresholds including persistence
       in time, timing, location and ice polygon size. In our approach we opted for fewer thresholds that can divide
       relatively large portions of ice, which makes some situations impossible to resolve. For instance, glacier ice inside
       the sea ice season is always classified as drift or landfast ice. Sea ice formation around water-cooling glacier ice
       is undetectable. The landfast ice edge fluctuation area is likely mapped as drift ice, because of alternating 'ice'

and 'open water' labels re-appearing in those pixels. However, most of these situations are challenging to resolve
       with available SAR imagery anyway, and an extensive field experiment is needed to better quantify those
       processes. Optical imagery can resolve some of these challenges (Makhotina et al., 2025), though the area is often
       cloud covered and the resolution needed to fully resolve such changes offered by, e.g., Planet Scope images are
       not regularly acquired over the area. Landsat and Sentinel-2 images are possible to use though the revisit times of

16 days (8 days if Landat-8 and -9 are combined) and 5 days respectively prevent from reliably resolve of landfast
       changes. Additionally, optical images do not cover the area in the polar night, and few are available from
       November to February, limiting the possibility of using optical images to capture the drift ice start up as well as
       the initial landfast ice formation period.

It is challenging to provide an accuracy measure for the applied method, particularly for potential errors from the
       binary classification that may propagate to the ice type mapping. Time series of ice type extent (Fig. 10) and maps
       of coverage (Fig. 11-12) appear reasonable, but were made with user-defined conditions. A small portion of ice
       extent (3.5%) was classified as 'masked' or 'uncertain'. This value can be used as an uncertainty envelope for the
       method, since some areas covered by these two classes can indeed include landfast, drift or glacier ice, while

others are likely speckle noise, radar shadow, tidal effect, etc. over ice-free water.

       The method will need calibration if used in other fjords with different size, shape, bathymetry and hydrography.
       Following are some particular features of Hornsund that dictated values of calibrated parameters. The cold
       Sørkapp Current precludes warm Atlantic Water entering the fjord and brings cold and fresh water masses as well

as pack ice from Storfjorden and the Barents Sea. This itself makes Hornsund hydrography very different from
       more northern fjords such as Bellsund or Isfjorden. The Barents Sea area experiences one of the highest sea ice
       declines in the Arctic which affects drift ice presence in the Hornsund area. Since the 'regime shift' in 2005, the
       pack ice at the entrance to the fjord has appeared episodically (Herman et al., 2025). On the other hand, the
       tidewater glaciers of Hornsund retreat is around twice as landfast as elsewhere in Svalbard (Błaszczyk et al.,

2013), has resulted in increasing fjord area (11% over 12 years; Table 6) and thus contributes glacier ice to the
       fjord. Retreating Hornbreen (Brepollen) and Hambergbreen (Storfjorden) are expected to cause the transition of
       Hornsund from a fjord to a straight in 2055-2065 (Grabiec et al., 2018).



Changes in the ice conditions impact the range of aspects of the fjord environments including coastal hazards.
Episodic presence of pack ice at the entrance to the fjord, coupled with increasing frequency and duration of
storms over the Greenland Sea (Wojtysiak et al., 2018), make more energetic waves enter the fjord more often.
Zagórski et al. (2015) observed an increased rate of shoreline retreat in Isbjørnhamna, the bay at which the PPS
Hornsund is located, in 1990-2011 relative to 1960-1990. The authors ascribed it to the intensification of storms
in the periods of positive air temperature anomalies and lack of sea ice cover. Our results suggest that the storm
waves occurring before January/February (average start of sea ice and landfast ice seasons) may have direct access
to the shore. However, Herman et al. (2025) observed the shift of maximal wave energy from late autumn
(November-December) in 1979-2005 to winter (December-March) in 2006-2023 caused by the decline in pack
ice. Increased nearshore wave energy translates into higher wave runup, and consequently a risk of coastal
flooding and erosion (Casas-Prat and Wang, 2020). These relationships are yet to be empirically found for
Hornsund. An important factor limiting wave action on the beach is ice present at the shore which in Hornsund
includes both glacier ice and shore ice (ice foot), a band of ice of diverse origin (glacier and sea ice, frozen swash
and splash, compacted snow) cemented to the shore (Rodzik and Zagórski, 2009). Our analyses did not include
ice on the shore due to the 100 m buffer along the coastline. Field observations suggest that shore ice is present
annually. It has recently rarely formed before January, though it persists until May-June. Future efforts should
include monitoring and modelling of nearshore wave transformation and wave energy delivery to the beach.

This study provides 11.5 years of near-daily maps of ice types which can be used to (i) assess the probability of
given ice conditions in specific time periods, e.g. for planning boat or snowmobile operations, and (ii) find a long-
term pattern in changing ice conditions in response to the climate change. However, we observe no interannual
trend in the length of the sea ice and landfast ice seasons or in the coverage of drift, landfast and glacier ice over
the study time period. We speculate that a long-term trend could be visible if the pre-2005 regime shift period was
incorporated (Muckenhuber et al., 2016; Herman et al., 2025). There is a great interannual variability in all ice
type extents and icy years occur both earlier (2012/13) and later (2019/20 and 2021/22) in the monitoring period.
Also seasonally, there is no shift in the peak extent of different ice types. Therefore, it appears that predictions of
ice conditions in a specific period can only be made by observing meteorological conditions leading to it, such as
autumn air or winter water temperature.

Pair-wise correlation between ice and environmental parameters shows a few interesting trends (Fig. 15). As the
date range of the end of sea ice season (same date as landfast ice season) is relatively limited (~1.5 months), the
length of the season is mostly dictated by the beginning of the season which ranges by up to 4 months for the sea
ice and 3 months for the landfast ice season. The strong relationship between the length of sea and landfast ice
seasons may suggest that earlier arrival of the pack ice from the Barents Sea causes an earlier formation of landfast
ice, which could imply one or both situations: the drift ice is anchored to the land forming a core of the landfast
ice or the presence of drift ice creates favourable conditions for the landfast ice to form, e.g. by decreasing water
temperatures or preventing waves from entering the inner parts of the fjord. The negative correlation between
winter water temperature and the length of ice season could suggest the pack ice lowering water temperature but
it may also mean the importance of local weather conditions since winter water temperature is dependent on



autumn air temperature (lag due to slower heat exchange of the water). If so, the monitoring of the air temperature at PPS Hornsund in autumn could allow an estimation of when the landfast ice forms.


The negative relationship between winter temperature and sea ice coverage is intuitive, but two interesting nuances should be noted. Firstly, is it air and not water temperature that controls the coverage which may be explained by the fact that further decrease of water temperature leads to more in situ freezing so water temperatures below - 1.9°C cannot be recorded and linked to the coverage. Secondly, stronger correlation characterises the coverage of
sea ice and landfast ice (on average 66% contributor to all sea ice) than drift ice which supports the hypothesis that drift ice is less dependent on local conditions than the landfast ice, being partly shaped by ice conditions in Eastern Svalbard as well as currents and winds outside Hornsund.

The negative correlation of glacier ice coverage and summer air and autumn water temperature supports that
during colder summers, ice produced from calving does not melt that quickly and remain on the fjord waters. The positive correlation of fjord area gain with autumn water temperature further points to the intensification of glacier calving in this period, while the negative correlation with spring air temperatures may show the periodic glacier front advance (Błaszczyk et al., 2013).

**6 Conclusions**

RADARSAT-2 imagery was used to extend an existing dataset of binary ice/open water maps in Hornsund fjord, Svalbard back in time. Subsequently, a 11.5-year archive of the binary maps at 50 m resolution was used to separate fjord ice into drift, landfast and glacier ice. The separation utilised a set of thresholds based on the familiarity with the study area and ice processes. The processing steps included (i) pixel-based splitting of the ice into stationary and moving classes using the 15-day persistence, (ii) masking radar shadows, (iii) polygon-based
classification into landfast, drift and glacier ice, and an uncertain ice class, and (iv) pixel-based refinement of drift and landfast ice.

In the result, a set of 2639 ice type maps was created for the 2012-02-01 to 2023-06-29 period with the average frequency of 1.59 days. These were used to characterize spatial and temporal (seasonal and interannual)
distribution of the ice types, and relate them to environmental conditions (air and water temperature, glacier front retreat).

Overall, 53% of the ice was classified as drift, 35% as landfast, 8.5% as glacier, 2.1% as masked and 1.4% as uncertain ice type. There was a great variability in the length of sea ice season (105-246 days) and landfast ice
season (61-195 days) as well as in the coverage of drift ice (9.3-26% fjord surface), landfast ice (3.1-40%) and glacier ice (3.7-5.2%) from year to year, with no clear long-term trend.

There was a negative correlation between the water temperature in winter months and the length of the sea ice and landfast ice season, as well as between the air temperature in winter months and sea ice and landfast ice
coverage. Glacier ice coverage depended on summer months air and autumn months water temperatures where



lower temperatures enhanced ice persistence. Fjord area gain was faster when autumns were warmer and springs were cooler.

The unprecedented dataset of near-daily high-resolution ice type maps over 11.5 years can be used in in a range of studies including modelling of fjord water circulation, wave transformation and runup, coastal flooding and erosion, and fjord ecology, as well as PPS management and boat/snowmobile transportation planning.

**Data availability**

The binary ice/open water maps are available in the PANGAEA repository; those based on Sentinel-1 are available at https://doi.pangaea.de/10.1594/PANGAEA.963167 (Swirad et al., 2023a) and those based on RADARSAT-2

are available at https://doi.pangaea.de/10.1594/PANGAEA.969031 (Swirad et al., 2024b). The maps of landfast, drift and glacier ice are available at https://doi.org/10.5281/zenodo.15647080 (Swirad, 2025).

**Supplement**

Table S1 in the Supplement is the time series of the extent ($m^2$) of the ice types in Hornsund.

**Author contributions**

ZMS and AMJ conceptualised the study. EM pre-processed the SAR scenes. ZMS developed the ice type separation method, processed and analysed the data with the help of AMJ. ZMS wrote the first draft. All authors edited the manuscript and agreed on its final version.

**Competing interests**

None of the authors has any competing interests.

**Acknowledgements**

We are grateful to Anthony Doulgeris (UiT) for the constructive discussion on image processing, and the Polar Polish Station Hornsund crew for maintaining the meteorological and oceanographic monitoring. Sentinel-1 imagery is freely available through the European Union's Earth observation Copernicus programme (https://copernicus.eu, last access: 21 July 2025). RADARSAT-2 data was provided by NCS/KSAT under the

Norwegian-Canadian RADARSAT-2 agreement 2011–16.

**Financial support**

This study was funded by the National Science Centre of Poland (SONATINA 5 grant no. 2021/40/C/ST10/00146) and the European Space Agency (HIRLOMAP grant no. 4000146036/24/I-DT-bgh). Zuzanna M. Swirad's visits to UiT were financed from the EEA and Norway Grants 2014-2021 operated by the





National Science Centre of Poland (CRIOS grant no. 2022/43/7/ST10/00001; HarSval grant no. 2023/43/7/ST10/00001). Eirik Malnes was partly supported by ESA DTC Svalbard and HarSval grants.

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
