# Peer review of "Distribution of landfast, drift and glacier ice in Hornsund, Svalbard"

_EGUsphere, 2025_

## Referee Comment (RC2)

**Distribution of landfast, drift and glacier ice in Hornsund, Svalbard - Review**

The authors present a study of fjord ice composition in Hornsund, Svalbard. An algorithmic approach to determining different ice classes is detailed and the resulting maps over 11.5 years of observations are investigated for trends and correlations. Remote sensing of fjord ice is rarely investigated and the paper presents novel results in that area of study. The language of the paper is clear and concise and most explanations are easy to follow.

**Minor Comments:**

**Specific:**

- 3.2.1: If I understand the method correctly, I think this paragraph should be slightly reworded so that it is more obvious from the start, that persistence refers to ice or water classes in the preliminary classification, not about the persistence of actual ice. I think it would help if terms such as 'moving' and 'stationary' pixels are avoided, as pixels do not move but rather the class in that pixel changes (or not).
- 3.2.3: It looks like the distance is arial not shortest path on water. This should probably be mentioned and potential errors discussed (1/2 sentences).
- 3.2.4 L. 299: "Time series of pixel values were used to re-classify drift ice into landfast ice in bays in the periods of high sea ice coverage over the entire fjord (Fig. 8e)"

  I do not understand this sentence. How were the time series used? Is this still automatic or is there manual inspection involved? Why is this necessary/correct?
- 5 L. 463: I am not sure the usage as an uncertainty 'envelope' is reasonable, as an envelope suggests a constraint which is not given here.
- 5 L. 524 onwards: "Secondly ...". I do not understand this sentence. Stronger correlation between which variables characterise the coverage?

**General:**

I think it would be good to discuss the potential impact of using drift retrievals to identify landfast ice rather than class persistence. As a reader I find it hard to judge how often the concentration of drift ice might be so high, that it is statistically likely to have a persistent ice pixel over 15 days, even though the ice is drifting (different ice in that pixel in each of the acquisitions).

It might be interesting to discuss if, based on your derived statistics, some reasonable predictions can be made for example in autumn about the length of the next ice seasons. Of course, 11.5 years is not a lot of data for such an endeavor, so one should still be careful with such statements.

As most of sea ice remote sensing relies on rather complex physics motivated or data driven per scene algorithms for delineating classes, I think it would be worth investing a paragraph into the advantages and disadvantages of forgoing this step and working with ice/water masks, thresholds and statistics instead. This will also establish this work more firmly in relation to existing scientific discourse.

**Technical/Language:**

- 41. "Lack of ..."
- 208. "same class (moving or stationary) were put ..." just recall what the classes are for clarity.
- Table 2. Please explain somewhere what marks the beginning and end of the sea ice/landfast ice season. Thank you.
- 272. Thresholds were set on which variables/parameters?
- 327. What are the integers in brackets? Are these numbers ever used again?
- 442. "...provided sufficient SNR values and good coherence between the two acquisitions ..."
- 442. "i.e. thickness dependent due to the level ice formation" I don't understand what is meant here, please clarify.
- 474. "...twice as landfast as...". This entire sentence needs to be restructured to be grammatically correct.
- 515. "... but it may also be a second order effect of local weather conditions..."
- 521. "... winter air temperature ..."
- 524. I think -1.9 should be unbreakable by linebreaks.
- 539. "... the 15-day persistence criterion ..."
- 543. "... period with **an** average frequency .... These were used to characterize **the** spatial and temporal ..."

---

## Author Response (AR1)

**RC1: Anonymous Referee #1**

General comments:

In "Distribution of landfast, drift and glacier ice in Hornsund, Svalbard" Swirad and coauthors undertook the task of splitting the fjord ice into different types based on maps covering years 2012 – 2023 and further relate properties of sea ice (the coverage, length of the sea ice season etc.) to local environmental factors to investigate seasonal and inter-annual trends. The manuscript is well organized and written with clearly defined goals, adequate methods, providing valuable and reasonable results. The paper represents substantial progress in sea ice investigation by developing an automated method for sea ice classification, thus extending beyond basic ice cover study. Resulting data are valuable and of great importance for broader scientific community and beyond.

However, there are some clarifications and improvements that need to be made before publication. Please find my comments below.

Specific comments:

Line 45: I'm not sure what do you mean by this sentence. I believe what you wanted to convey from Styszyńska and Rozwadowska (2008) is that in situ sea ice is formed when fjord's water reaches freezing temperature. What about brine release during ice formation?

We have clarified the phrase to

"*Some examples of the interactions between glacier ice and sea ice include in situ sea ice formation around glacier ice where surface water is cooled [...]*".

We do not mention brine release here because we strictly refer to glacier ice-sea ice interaction.

The last paragraph of Introduction states the goals of the paper. What I lack here is a clear statement about novelty that this study provides. Beyond extending time series from Swirad et al. (2023a), enormous workload and effort has been devoted to classifying/splitting the fjord ice into different types, among other things, by developing an automated method.

We have modified the paragraph to:

*"The main goal of this study was to develop an automated method to map landfast, drift and glacier ice in fjord environments. To do so, we extended near-daily high-resolution binary ice/open water maps for Hornsund, Svalbard (Swirad et al., 2023a) back in time to include three more sea ice seasons (Swirad et al, 2024b). We then used a set of calibrated thresholds to split the fjord ice into ice types. The maps cover the southern part of Svalbard around the Polish Polar Station (PPS) Hornsund and span over 11.5 seasonal cycles, 01.2012-06.2023. The length of the dataset combined with extensive in situ measurements of air and water temperature enabled us to analyse seasonal and intern-annual trends and relate them to local climatological changes."*

Lines 115-116: Was the buffer extended for the DN area as in previous study (Swirad et al. (2024))?

Yes, we have clarified it:

*"Additionally, a 100 m buffer was added along the coast to exclude the tidal zone, with a larger buffer area in the shallow, deltaic Dunøysundet/Nottinghambukta (DN) area (for location see Fig. 1)."*

Section 3.2.1 I do not follow the initial division between stationary and moving ice. For example, within 5 days, how exactly is the stationary ice and moving ice defined?

The issues with section 3.2.1 were raised by both reviewers and in the community comment. We re-wrote the paragraph for clarity.

*"The persistence of ice in single locations (pixels) over consecutive days was first determined under the assumption that landfast ice persists in the same locations longer than drift and glacier ice. The aim was to divide ice into 'stationary' and 'moving' based on uninterrupted occurrence in the same locations over a certain number of consecutive days. If the ice was present in a pixel on every day over these periods, it was classified as 'stationary' ice. Otherwise, it was classified as 'moving' ice. To optimize the number of days for the separation, a 13-month subset of maps (July 2018 - July 2019; n = 293) was analysed. The total area of stationary and moving ice was extracted for bays and open Greenland Sea (as delimited in Fig. 1) based on an uninterrupted persistence during 5, 10, 15 and 20 subsequent days. Fig. 4 shows that in the bays the area of moving ice was dynamic throughout the year likely reflecting the glacier and drift ice fluxes as well as landfast ice edge fluctuations."*

Section 5. I have the most questions about discussion, in particular pair-wise correlation. As it was shown, sea ice formation and evolution is a complex process, dependent on many factors. First of all, it would be useful to give which correlations are statistically significant. This would show exactly which factors, among so many, should be the focus of this part of the discussion.

We marked the statistically insignificant relationships in Figure 15, and throughout the text state the p-value next to the PCC value. The key relationships that we describe are all statistically significant ($p < 0.05$).

Secondly, is salinity data available? I think, it's an important factor, when we talk about sea ice. Briefly, the sea water must cool to freezing point. How fast it happens depends on water salinity. The higher initial salinity (at the end of summer season or in autumn) the lower air temperature and sufficiently long winter is required to cool the water and initiate the sea ice formation. This would partly (excluding further modification by e.g. wind) control properties of landfast ice. Deeper, more complex discussion is needed to explain existing relationships.

We have added a paragraph in the discussion that tackles this issue:

*"Here we only investigate water temperature at the three-month (season) scale. However, the in situ sea ice formation by water freezing is a complex process where not only temperature but also salinity, wind conditions and sea state are important (Lubin and Massom, 2006; Wang et al., 2021). The freezing point of seawater decreases with increasing salinity (Roquet et al., 2022). Thus, saline Atlantic Water requires stronger cooling to reach freezing compared to fresher Arctic Water. Consequently, the inflow of saline Atlantic Water versus fresh Arctic Water into the fjord in summer/autumn may be critical for the formation of landfast ice in winter (Promińska et al., 2018). Moreover, during freeze-up, salt is expelled from the forming ice (brine rejection), enhancing the stratification of the water column with*

*relatively fresh water accumulating near the surface and more saline water at depth. Continued hydrographic monitoring and higher-resolution observations of ice formation than those presented in this study are essential to improve our understanding of sea ice–ocean interactions in Hornsund."*

Technical corrections:

Line 60: change intern-annual to inter-annual

Changed to 'inter-annual'.

Line 336: change "in the deepest parts" to "the most inner parts"

Changed to 'inner-most'.

Line 350: Does this apply to the landfast ice season or to the sea ice season as a whole. From Fig. 10 I can see that the iciest seasons are 2012/2013, 2014/2015, 2019/2020 and 2021/2022? 2012/2013 is even more icy than two seasons mentioned in the text.

Thank you. We have clarified that it refers to the entire sea ice season and added the other two icy seasons as suggested:

*"The least icy sea ice seasons were 2013/14, 2011/12 and 2015/16, while the iciest were 2012/13, 2019/20 and 2021/22 and 2014/15 (Fig. 10)."*

Line 355: I would include season 2014/2015 to those with the highest coverage as well. It is also clearly seen in Figure 11.

The 2014/15 was added as suggested.

Lines 387-388: For clarity please add "mean autumn water temperatures"

The word 'water' was added for clarity.

Line 474: please correct "twice as landfast to" twice as fast"

The word was changed from 'landfast' to 'fast'.

Line 475: Fjord area gain is shown in Table 7

The reference to Table 7 was put instead of Table 6.

Line 521: is it winter air or water temperature?

The word 'air' was added for clarity.

Figure 7. Please consider increasing font size. They are barely legible.

*The figure was change from horizontally- to vertically-aligned panels which allowed increasing the font.*

Figure 9. In the caption: there is "a) landfast ice, b) drift ice", according to the Figure it should be "a) drift ice, b) landfast ice". Throughout the manuscript you write about landfast ice and in Figure b) there is a Fast ice. Please consider changes in the figure or just add in caption b) landfast ice (Fast ice), also maybe it's worth to add in line 27 (in Introduction) "Landfast ice (interchangeably called Fast ice)".

*The caption has been modified to reflect the panel order. 'Fast' was changed to 'landfast' for consistency.*

Figure 10. Please consider increasing font size. Figure 8 is a good example of it.

*The font was increased as suggested.*

Figure 13: In the caption you refer to panels marked a, b and c but the letters are missing in the Figure. For unification, please change "fast" in the middle panel to "landfast".

*The panel letters have been added, and the title have been removed.*
* * *
**RC2: Karl Kortum**

The authors present a study of fjord ice composition in Hornsund, Svalbard. An algorithmic approach to determining different ice classes is detailed and the resulting maps over 11.5 years of observations are investigated for trends and correlations. Remote sensing of fjord ice is rarely investigated and the paper presents novel results in that area of study. The language of the paper is clear and concise and most explanations are easy to follow.

Minor Comments:

Specific:

3.2.1: If I understand the method correctly, I think this paragraph should be slightly reworded so that it is more obvious from the start, that persistence refers to ice or water classes in the preliminary classification, not about the persistence of actual ice. I think it would help if terms such as 'moving' and 'stationary' pixels are avoided, as pixels do not move but rather the class in that pixel changes (or not).

*See the similar comment by Reviewer #1. We re-wrote the paragraph for clarity.*

*"The persistence of ice in single locations (pixels) over consecutive days was first determined under the assumption that landfast ice persists in the same locations longer than drift and glacier ice. The aim was to divide ice into 'stationary' and 'moving' based on uninterrupted occurrence in the same locations over a certain number of consecutive days. If the ice was present in a pixel on every day over these periods, it was classified as 'stationary' ice. Otherwise, it was classified as 'moving' ice. To optimize the number of days for the separation, a 13-month subset of maps (July 2018 - July 2019; n = 293) was analysed. The total area of stationary and moving ice was extracted for bays and open Greenland Sea (as delimited in Fig. 1) based on an uninterrupted persistence during 5, 10, 15 and 20 subsequent days. Fig. 4 shows that in the bays the area of moving ice was dynamic throughout the year likely reflecting the glacier and drift ice fluxes as well as landfast ice edge fluctuations."*

3.2.3: It looks like the distance is arial not shortest path on water. This should probably be mentioned and potential errors discussed (1/2 sentences).

*It was clarified at the beginning of the section and in Figure 6 caption:*

*"The annually-updated (July-June) maps of aerial distance from land, bays and glacier fronts, and the single maps of aerial distance from the open sea and to Dunøysundet/Nottinghambukta (DN) area [...]".*

*"Figure 6. Maps of the aerial distance [...]"*

*We have added a note on potential errors in the discussion:*

*"For instance, using aerial distances from bays, sea, land, glacier fronts and DN area rather than shortest path on water was sufficient in Hornsund given its morphology but may be erroneous in different fjords."*

3.2.4 L. 299: "Time series of pixel values were used to re-classify drift ice into landfast ice in bays in the periods of high sea ice coverage over the entire fjord (Fig. 8e)" I do not understand this sentence. How were the time series used? Is this still automatic or is there manual inspection involved? Why is this necessary/correct?

*This step is the response to the problem raised in the previous section which reads "During the cold seasons the temporal constraint still allowed the landfast ice to be present in the main basin and the open sea (E in Fig. 8b). The spatial constraint limited this effect but resulted in the entire polygon transition to the drift ice class (F in Fig. 8c). This was addressed later at the pixel level (last step in Fig. 3)."*

*We have modified section 3.2.4 for clarity:*

*"Pixel values through time were used to automatically re-classify drift ice into landfast ice in bays in the periods of high sea ice coverage over the entire fjord (Fig. 8e). This means that if a pixel was ascribed 'landfast ice' class, then 'drift ice' and then 'landfast ice' again, it was considered 'landfast' throughout."*

5 L. 463: I am not sure the usage as an uncertainty 'envelope' is reasonable, as an envelope suggests a constraint which is not given here.

*We have replaced 'envelope' with 'measure'.*

5 L. 524 onwards: "Secondly …". I do not understand this sentence. Stronger correlation between which variables characterise the coverage?

*We have modified the sentence for clarity:*

*"Secondly, stronger correlation exists between winter air temperature and the sea ice and landfast ice coverage (on average 66% contributor to all sea ice) than between winter air temperature and drift ice which supports [...]"*

General:

I think it would be good to discuss the potential impact of using drift retrievals to identify landfast ice rather than class persistence. As a reader I find it hard to judge how often the concentration of drift ice might be so high, that it is statistically likely to have a persistent ice pixel over 15 days, even though the ice is drifting (different ice in that pixel in each of the acquisitions).

Thank you for the comment. We added a paragraph on the drift retrieval:

*"Drift ice algorithms often rely on feature tracking (Muckenhuber and Sandven, 2017; Korosov and Rampal, 2017; Gao et al., 2025) to determine the drift between two subsequent SAR images. The landfast ice formed within the study area typically has a low degree of textural features, and the drift ice originating from the landfast ice likely has floe size smaller than the pixel size (50 m) and as such using drift retrievals for the ice formed within the fjord is challenging. Drift retrievals may be used to capture the ice entering the fjord system from the Greenland Sea, though the method may be challenging if the drift ice gets incorporated into the landfast ice areas."*

It might be interesting to discuss if, based on your derived statistics, some reasonable predictions can be made for example in autumn about the length of the next ice seasons. Of course, 11.5 years is not a lot of data for such an endeavor, so one should still be careful with such statements.

We mention it twice in the discussion:

*"Therefore, it appears that predictions of ice conditions in a specific period can only be made by observing meteorological conditions leading to it, such as autumn air or winter water temperature."*

*"The statistically significant negative correlation between winter water temperature and the length of ice season could suggest the pack ice lowering water temperature but it may also be a second order effect of local weather conditions since winter water temperature is dependent on autumn air temperature (lag due to slower heat exchange of the water). If so, the monitoring of the air temperature at PPS Hornsund in autumn could allow an estimation of when the landfast ice forms."*

As most of sea ice remote sensing relies on rather complex physics motivated or data driven per scene algorithms for delineating classes, I think it would be worth investing a paragraph into the advantages and disadvantages of forgoing this step and working with ice/water masks, thresholds and statistics instead. This will also establish this work more firmly in relation to existing scientific discourse.

We have added a paragraph to the discussion;

*"The sea ice within the Hornsund basin is predominantly first-year ice that forms under calm conditions as landfast sea ice, and as such has an inherently low textural variability (Kim et al., 2020) and for part of the season has low backscatter due to the new ice formation stage (Liu et al., 2025). Textural information is an integral part in many existing sea ice type classification methods (Park et al., 2020), and level landfast ice is known to be challenging to accurately classify with existing sea ice classification methods (Kim et al., 2020; Wang et al., 2023; Zhu et al., 2025). Hence, inherent ambiguities in the SAR images between different ice types may transfer inaccuracies in sea ice type classification to the accuracy in separation*

*between sea ice and open water. Although they are similar in backscatter values, the types of ice can be separable by their persistence through time, location, timing, size and shape."*

Technical/Language:

41. "Lack of …"

Modified as suggested.

208. "same class (moving or stationary) were put …" – just recall what the classes are for clarity.

It was modified for clarity:

*"The masked moving and stationary ice maps were polygonised in such a way that all touching pixels of the same class (moving/stationary) were put into the same polygon. The polygons were characterised by a number of parameters: ice class (moving/stationary), date […]"*

Table 2. Please explain somewhere what marks the beginning and end of the sea ice/landfast ice season. Thank you.

We have clarified it in the Table 2 caption:

*"The dates of onset and end of drift and landfast ice were defined visually from SAR imagery. The sea ice season is a period between the day with first drift ice inside the fjord and the landfast ice end, and the landfast ice season is a period between the onset and the end of landfast ice (Swirad et al., 2024a)."*

272. Thresholds were set on which variables/parameters?

The thresholds are described in this and next paragraph. We have modified the sentence for clarity:

*"Subsequently, a set of thresholds described below was set to split the moving ice into drift ice and glacier ice."*

327. What are the integers in brackets? Are these numbers ever used again?

The integers were removed.

442. "…provided sufficient SNR values and good coherence between the two acquisitions …"

Modified to:

*"Moreover, single-pass InSAR can be used to identify the landfast ice extent provided high interferometric coherence and sufficient SNR values which partially depends on the ice thickness due to the low SNR in early ice formation stages and low backscatter values for level ice with low amount of textural features (Dierking et al., 2017)."*

442. " i.e. thickness dependent due to the level ice formation" – I don't understand what is meant here, please clarify.

Clarified:

"*Moreover, InSAR can be used to identify the landfast ice extent provided high interferometric coherence and sufficient SNR values which depend on the ice thickness due to the level ice formation (Dierking et al., 2017).*"

474. "…twice as landfast as…". This entire sentence needs to be restructured to be grammatically correct.

Modified to "twice as fast as".

515. "… but it may also be a second order effect of local weather conditions…"

Modified as suggested.

521. "… winter air temperature …"

Modified as suggested.

524. I think -1.9 should be unbreakable by linebreaks.

Imposed no breaking hyphen (ctrl shift -) in Word.

539. "… the 15-day persistence criterion …"

Modified as suggested.

543. "… period with an average frequency …. These were used to characterize the spatial and temporal …"

Modified as suggested.
* * *
**CC1: Wang Zihan**

This study provides valuable data and an interesting perspective on fjord ice and its environmental drivers. I do, however, have a few suggestions that might strengthen the manuscript.

First, the classification of pixels into "stationary" and "moving" is central for distinguishing fast ice, but the description is not sufficiently clear. The manuscript states that "The ice pixels were first divided into 'stationary' or 'moving' classes depending on their persistence through time", but does not explain how this was actually implemented in practice. It seems possible that a feature-tracking approach was applied, but this is not stated. Clarification would improve transparency and reproducibility.

The issues with section 3.2.1 were raised by both reviewers and in the community comment. We re-wrote the paragraph for clarity.

*"The persistence of ice in single locations (pixels) over consecutive days was first determined under the assumption that landfast ice persists in the same locations longer than drift and glacier ice. The aim was to divide ice into 'stationary' and 'moving' based on uninterrupted occurrence in the same locations over a certain number of consecutive days. If the ice was present in a pixel on every day over these periods, it was classified as 'stationary' ice. Otherwise, it was classified as 'moving' ice. To optimize the number of days for the separation, a 13-month subset of maps (July 2018 - July 2019; n = 293) was analysed. The total area of stationary and moving ice was extracted for bays and open Greenland Sea (as delimited in Fig. 1) based on an uninterrupted persistence during 5, 10, 15 and 20 subsequent days. Fig. 4 shows that in the bays the area of moving ice was dynamic throughout the year likely reflecting the glacier and drift ice fluxes as well as landfast ice edge fluctuations."*

The sentence in the abstract was also modified to:

*"The ice was first divided into stationary and moving classes based on persistence in space (pixels) through time."*

Second, the manuscript presents correlations between fjord ice parameters, air temperature, and water temperature, but does not provide p-values. Since the time series are relatively short, especially for water temperature where gaps further reduce the effective sample size, it is difficult to know whether the reported Pearson coefficients are statistically significant or could arise from chance. This is particularly important because the conclusions rely heavily on these correlations to link environmental drivers with ice variability. Including p-values would make the statistical basis of these conclusions much stronger.

We marked the statistically insignificant relationships in Figure 15, and throughout the text state the p-value next to the PCC value. The key relationships that we describe are all statistically significant (p < 0.05).

Finally, the statement that no interannual trend was observed is currently descriptive. I suggest that the authors quantify the trends in the key time series using a Mann–Kendall test combined with Sen's slope, and report both the trend values and their p-values, regardless of whether they are significant. This would provide a clearer and more transparent basis for the conclusion on long-term changes.

Thank you for this comment. We have added the Mann-Kendall test to section 4. We did not calculate Sen's slope because MK test showed that all trends were insignificant.

*"The Mann-Kendall test (Mann, 1945; Kendall, 1975) was used to verify whether an interannual trend existed in season duration and ice coverage over the analysed 11.5 years. For all parameters the trend was positive and insignificant with the τ and p values of 0.09 and 0.73 for the length of sea ice season, 0.17 and 0.49 for the length of landfast ice season, 0.06 and 0.84 for the drift ice coverage during the sea ice season, and 0.27 and 0.24 for the landfast ice coverage during the landfast ice season."*

---

## Referee Report (RR1)

Thank you for your explanations and the clarifications and changes made to the manuscript. I just have some short remarks about two of the responses.

i. 5 L. 524 onwards: "Secondly …". I do not understand this sentence. Stronger correlation between which variables characterise the coverage?

We have modified the sentence for clarity:
"Secondly, stronger correlation exists between winter air temperature and the sea ice and landfast ice coverage (on average 66% contributor to all sea ice) than between winter air temperature and drift ice which supports […]"

*I am afraid I still not sure that I understand this wording. To me it reads like this: "correlation (A, B, C) > correlation (A, D)", and I don't understand what the correlation between three properties is.*

ii. It might be interesting to discuss if, based on your derived statistics, some reasonable predictions can be made for example in autumn about the length of the next ice seasons. Of course, 11.5 years is not a lot of data for such an endeavor, so one should still be careful with such statements.

We mention it twice in the discussion:

"Therefore, it appears that predictions of ice conditions in a specific period can only be made by observing meteorological conditions leading to it, such as autumn air or winter water temperature."

"The statistically significant negative correlation between winter water temperature and the length of ice season could suggest the pack ice lowering water temperature but it may also be a second order effect of local weather conditions since winter water temperature is dependent on autumn air temperature (lag due to slower heat exchange of the water). If so, the monitoring of the air temperature at PPS Hornsund in autumn could allow an estimation of when the landfast ice forms."

*Thank you for pointing out the respective sections. I realise, I have not worded my original comment clearly. I was actually wondering if it might be interesting to train a small predictor (linear regression is sufficient) based on your derived variables and discuss the results. I do think the paper is complete without it, I just thought it might be an interesting way to show off your results.*

---

## Author Response (AR2)

**Referee #1**

I would like to thank the authors for improving the manuscript and addressing all my concerns. I have only two minor suggestions:

Line 566: I think a suitable reference here would be Arntsen et al., (2019). Inflow of Warm Water to the Inner Hornsund Fjord, Svalbard: Exchange Mechanisms and Influence on Local Sea Ice Cover and Glacier Front Melting, as the authors discuss exactly how different hydrographic conditions influence sea-ice cover.

We have replaced the reference as suggested.

Lines 567 - 569: I suggest to remove this sentence, as it is not entirely accurate. Brine release during ice formation drives vertical convection and mixing, which reduces stratification and leads to deepening of the winter mixed layer.

We have removed the sentence as suggested.

**Referee #2: Karl Kortum**

Thank you for your explanations and the clarifications and changes made to the manuscript. I just have some short remarks about two of the responses.

L. 524 onwards: "Secondly …". I do not understand this sentence. Stronger correlation between which variables characterise the coverage?

*We have modified the sentence for clarity:*

*"Secondly, stronger correlation exists between winter air temperature and the sea ice and landfast ice coverage (on average 66% contributor to all sea ice) than between winter air temperature and drift ice which supports […]"*

I am afraid I still not sure that I understand this wording. To me it reads like this: "correlation (A, B, C) > correlation (A, D)", and I don't understand what the correlation between three properties is.

Thanks for the comment, we have further clarified it to:

*"Secondly, stronger correlation exists between winter air temperature and the sea ice coverage, and between winter air temperature and the landfast ice coverage (on average 66% contributor to all sea ice) than between winter air temperature and the drift ice coverage which supports […]"*

It might be interesting to discuss if, based on your derived statistics, some reasonable predictions can be made for example in autumn about the length of the next ice seasons. Of course, 11.5 years is not a lot of data for such an endeavor, so one should still be careful with such statements.

*We mention it twice in the discussion:*

*"Therefore, it appears that predictions of ice conditions in a specific period can only be made by observing meteorological conditions leading to it, such as autumn air or winter water temperature."*

*"The statistically significant negative correlation between winter water temperature and the length of ice season could suggest the pack ice lowering water temperature but it may also be a second order*

*effect of local weather conditions since winter water temperature is dependent on autumn air temperature (lag due to slower heat exchange of the water). If so, the monitoring of the air temperature at PPS Hornsund in autumn could allow an estimation of when the landfast ice forms."*

Thank you for pointing out the respective sections. I realise, I have not worded my original comment clearly. I was actually wondering if it might be interesting to train a small predictor (linear regression is sufficient) based on your derived variables and discuss the results. I do think the paper is complete without it, I just thought it might be an interesting way to show off your results.

Thank you. We have added a sentence in the discussion:

*"Our data suggest that a 1°C decrease in the average October-December air temperature results in 19-day longer sea ice season (linear regression $R^2$ = 0.53)."*